# No current evidence for widespread dosage compensation in *S. cerevisiae*

Eduardo M Torres[1]*, Michael Springer[2]*[†], Angelika Amon[3,4,5]*[†]

[1]Department of Molecular, Cell and Cancer Biology, University of Massachusetts Medical School, Worcester, United States; [2]Department of Systems Biology, Harvard Medical School, Boston, United States; [3]David H. Koch Institute for Integrative Cancer Research, Massachusetts Institute of Technology, Cambridge, United States; [4]Department of Biology, Massachusetts Institute of Technology, Cambridge, United States; [5]Howard Hughes Medical Institute, Massachusetts Institute of Technology, Cambridge, United States

**Abstract** Previous studies of laboratory strains of budding yeast had shown that when gene copy number is altered experimentally, RNA levels generally scale accordingly. This is true when the copy number of individual genes or entire chromosomes is altered. In a recent study, Hose et al. (2015) reported that this tight correlation between gene copy number and RNA levels is not observed in recently isolated wild *Saccharomyces cerevisiae* variants. To understand the origins of this proposed difference in gene expression regulation between natural variants and laboratory strains of *S. cerevisiae*, we evaluated the karyotype and gene expression studies performed by Hose et al. on wild *S. cerevisiae* strains. In contrast to the results of Hose et al., our reexamination of their data revealed a tight correlation between gene copy number and gene expression. We conclude that widespread dosage compensation occurs neither in laboratory strains nor in natural variants of *S. cerevisiae*.

*For correspondence: eduardo. torres@umassmed.edu (EMT); michael_springer@hms.harvard. edu (MS); angelika@mit.edu (AA)

[†]These authors contributed equally to this work

Competing interests: The authors declare that no competing interests exist.

## Introduction

Losses or gains of whole chromosomes, a condition known as aneuploidy, have a profound impact on cell physiology. Gene expression studies in budding yeast, fission yeast, mammalian cells, and plants revealed that this is due to the fact that changes in gene copy number result in changes in gene expression (*Chikashige et al., 2007*; *Huettel et al., 2008*; *Pavelka et al., 2010*; *Sheltzer et al., 2012*; *Stingele et al., 2012*; *Torres et al., 2010*; *2007*). For example, in haploid budding yeast strains harboring single chromosome gains, RNA levels of more than 90% of genes located on the extra chromosome reflect the increased gene copy number (*Dephoure et al., 2014*; *Torres et al., 2007*). Only few genes, such as histone and some ribosomal genes defy this trend (*Dabeva and Warner, 1987*; *Gunjan and Verreault, 2003*; *Libuda and Winston, 2006*; *Moran et al., 1990*; *Sutton et al., 2001*; *Vilardell and Warner, 1997*). Given that aneuploidy has such a profound impact on the cell's transcriptome and proteome it is not surprising that aneuploidy affects virtually all aspects of cell physiology, generally having a negative impact on fitness (*Hassold and Jacobs, 1984*; *Hodgkin, 2005*; *Huettel et al., 2008*; *Lindsley et al., 1972*; *Niwa et al., 2006*; *Stingele et al., 2012*; *Torres et al., 2007*; *Williams et al., 2008*).

Aneuploidy not only affects gene expression through changes in gene copy number, the condition also causes transcriptional responses. For example, when chromosome gains or losses lead to a decrease in growth rate, a stereotypic slow-growth transcriptional response known as the environmental stress response (ESR) ensues (*Gasch et al., 2000*). The ESR is characterized by the down-regulation of growth-promoting genes and the up-regulation of stress response genes and has been

**eLife digest** DNA inside cells is packaged into structures called chromosomes. Different species can have different numbers of chromosomes, but when any cell divides it must allocate the right number of chromosomes to each new cell. If this process goes wrong, cells end up with too many or too few chromosomes. The presence of extra copies of the genes on the additional chromosomes can cause the levels of the proteins encoded by those genes to rise abnormally, which can in turn lead to cell damage and disease.

Proteins are produced using the information in genes via a two-step process. First, the gene's DNA is copied to create molecules of RNA, and these molecules are then translated into proteins. In many organisms, the presence of extra chromosomes in a cell is matched by a corresponding increase in the RNA molecules encoded by the extra genes. Some organisms, however, counteract this effect through a process called dosage compensation. This process inactivates single genes or whole chromosomes by various means, and ensures that normal levels of RNA are produced, even in the presence of extra genes.

In 2015, researchers from the University of Wisconsin-Madison reported that dosage compensation occurs in wild strains of budding yeast and effectively protects the yeast cells against the harmful effects of having extra chromosomes. However, these findings conflicted with earlier studies of laboratory strains of this yeast, which had reported that RNA levels increased along with gene number.

Torres, Springer and Amon have re-analysed the data published in 2015, and now challenge the findings of the previous study involving the wild yeast strains. The new re-analysis instead showed that, like in laboratory yeast strains, gene number still correlates closely with RNA levels in the wild yeast. This led Torres, Springer and Amon to conclude that, in contrast with the previous report, there is currently no evidence that dosage compensation occurs in wild strains of yeast.

So why do the results of these two studies disagree? Torres, Springer and Amon identified several issues concerning the original analysis made by the researchers from the University of Wisconsin-Madison. For example, some of the strains included in the 2015 study were unstable and were naturally losing the additional chromosomes that they'd acquired. Also, the thresholds set in the analysis to identify dosage compensated genes do not appear to have been stringent enough. Together, the new findings indicate that dosage compensation is a rare event in both wild and laboratory strains of yeast.

reported to occur in response to aneuploidy in many organisms including laboratory yeast strains (*Sheltzer et al., 2012*).

Changes in gene copy number not only can lead to transcriptional responses but also can elicit dosage compensation, a gene regulatory mechanism that specifically compensates for alterations in gene copy number at the gene expression level. Dosage compensation is best understood in the context of sex chromosome-encoded genes (reviewed in *Straub and Becker, 2007*). For example in mammals, an RNA-mediated mechanism silences expression of one copy of the X chromosome in females thereby equalizing X chromosome-encoded gene expression between males and females (*Lee and Bartolomei, 2013*). In *Caenorhabditis elegans*, gene expression of the two X chromosomes is reduced by half in the hermaphrodite to match the expression of the single X chromosome in males (*Meyer, 2010*). Dosage compensation can also affect specific loci. The perhaps best known example is the histone locus in budding yeast (*Osley and Hereford, 1981*). When an extra copy of the *HTA1* gene (histone H2A) is introduced into budding yeast, mRNA turnover increases resulting in normal *HTA1* transcript levels (*Moran et al., 1990*; *Osley and Hereford, 1981*). It is important to note that dosage compensation and transcriptional responses to aneuploidy can have the same effect on a gene. Both can cause the down-regulation of a gene, but the mechanisms are distinct. Transcriptional responses to aneuploidy are elicited by an aneuploid genome affecting a biological pathway and are not restricted to the aneuploid chromosomes but impact expression of genes located throughout the genome. In contrast, dosage compensation specifically alters the expression

of a gene whose copy number has been varied and its effects are thus restricted to the aneuploid chromosome.

Experimental evolution studies suggest that selective pressures cause changes in karyotype such as chromosome gains or losses (*Dunham et al., 2002*; *Gresham et al., 2008*). However, such aneuploidies are usually transient evolutionary intermediates that, given time, are replaced with more optimal solutions (*Yona et al., 2012*). A key question that arises from these studies is how prevalent whole chromosome gains and losses are in wild yeast strains and how aneuploidies affect cell physiology. *Hose et al. (2015)* addressed these questions. They isolated 47 wild yeast strains to identify 12 (26%) that harbored whole chromosome gains and/or losses. The detailed analysis of six of these strains led them to the conclusion that aneuploidies are prevalent, stable and well-tolerated in wild yeast strains. Based on gene expression analyses, they further concluded that tolerance to aneuploidy is caused by dosage compensation mechanisms that buffer gene amplifications thereby protecting cells against the adverse effects of aneuploidy. They reported that gene-dosage compensation functions at >30% of amplified genes.

To understand why dosage compensation mechanisms are rare in laboratory strains of budding yeast, but highly prevalent in wild isolates, we reevaluated the karyotype and gene expression studies performed by *Hose et al. (2015)*. This reexamination revealed that gene copy number and expression are tightly correlated in wild *S. cerevisiae* strains. We conclude that dosage compensation is a rare occurrence in both, laboratory and natural variants of *S. cerevisiae*.

## Results

### Many wild yeast strains have heterogeneous karyotypes

*Hose et al. (2015)* isolated 47 wild yeast variants and determined their karyotypes by inferring the copy number from genome sequencing data using depth of coverage. This analysis showed that 12 of these 47 strains harbor whole chromosome aneuploidies. DNA and RNA sequencing data for 6 of these 12 aneuploid strains were deposited in the NCBI Sequence Read Archive (SRA) under accession SRP047341 and NCBI Gene Omnibus under accession GSE61532 referenced in *Hose et al. (2015)*. Three of these strains harbored one or two single chromosome gains in a diploid background. Strain K9 is a diploid strain carrying an extra copy of chromosomes IX and X (*Figure 1A,G*), YPS1009 is diploid with an extra copy of chromosome XII (*Figure 1B,G*), and diploid strain NCYC110 carries two extra copies of chromosome VIII (*Figure 1C,G*). In addition, *Hose et al. (2015)* analyzed three strains with high levels of aneuploidy. These strains were strains YJM428, Y2189 and K1 (*Figure 1D–G*).

We examined the karyotypes and gene expression of these strains and found the aneuploid strains K9, YPS1009 and NCYC110 with low levels of aneuploidy to harbor relatively stable karyotypes (*Figure 1A–C*). As discussed in more detail below, RNA levels also generally correlated well with DNA levels, with aneuploid chromosomes overall showing a corresponding increase in gene expression (*Figure 1A–C*). It is, however, noteworthy that strain K9 which harbors extra copies of chromosome IX and X in the *Hose et al. (2015)* study was previously reported to be trisomic for chromosome IX only, indicating that this strain exhibits some karyotypic instability (*Kvitek et al., 2008*).

In contrast to the relatively stable strains K9, YPS1009, and NCYC110, a different picture emerged from our analysis of strains YJM428, Y2189, and K1 that harbor complex karyotypes. Based on the presence of non-integer DNA copy number states, we conclude that the described aneuploidies are only present in subpopulations of cells (*Figure 1D–F*). The comparison between RNA and DNA levels further revealed significant inconsistencies between the two data sets indicating that some strains had changed their karyotypes between the two analyses (e.g. DNA and RNA copy numbers are very different in strains Y2189 and K1; *Figure 1E,F*). This discrepancy is problematic as *Hose et al. (2015)* used the standard deviations (SD) of the DNA measurements to establish cutoffs in their RNA data set to identify dosage compensated genes (discussed in detail below).

We also analyzed the karyotypes of the other six aneuploid variants UC5, WE372, T73, Y3, Y6, and CBS7960 that were not characterized in detail by Hose and coworkers (both Figure 1 and Supplementary file 1 in *Hose et al., 2015*; $\log_2$ ratios of normalized DNA copy numbers were provided by A. Gasch). We found that strains T73, which is tetrasomic for chromosome VIII (analyzed below;

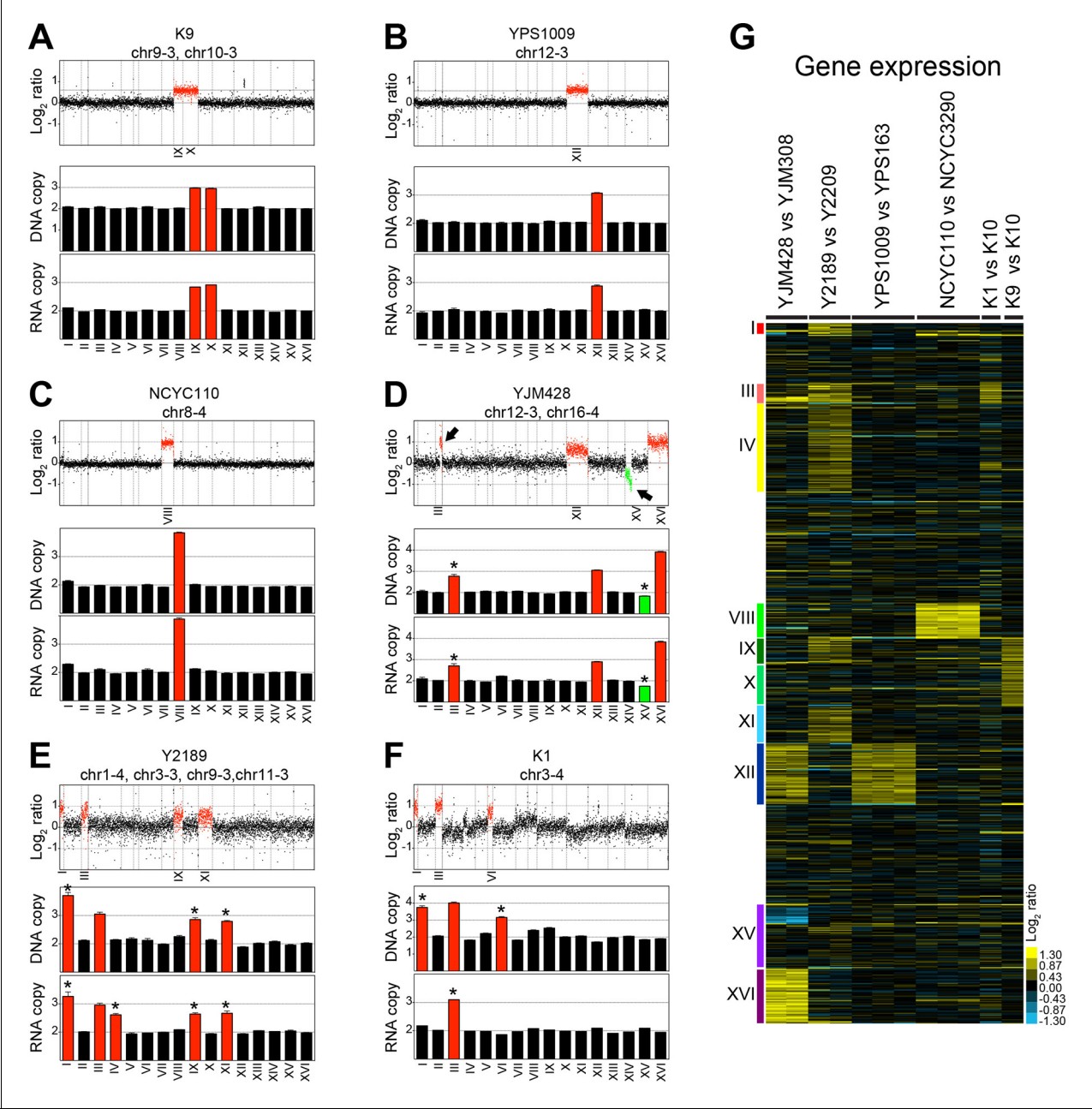

**Figure 1.** DNA and RNA copy number of six wild *S. cerevisiae* strains. (**A**) DNA and RNA copy number analysis of strain K9 compared to K10. Log$_2$ ratios of aneuploid vs. euploid DNA in the order of the chromosomal location of their encoding genes are shown on the top. DNA copy number of chromosomes IX and X are shown in red. The graph below shows the average DNA copy number per chromosome. The graph below shows RNA copy number averaged per chromosome relative to K10 (n = 1). (**B**) DNA and RNA copy number analysis of strain YPS1009 compared to YPS163. Data are represented as in (**A**). Error bars represent the SD of the chromosome means from three biological replicates. Medians are identical to the means. (**C**) DNA and RNA copy number analysis of strain NCYC110 compared to NCYC3290. Data are represented as in (**A**). Error bars represent the SD of the chromosome means from three biological replicates. Medians are identical to the means. (**D**) DNA and RNA copy number analysis of strain YJM428 compared to YJM308. Log$_2$ ratios of aneuploid vs. euploid DNA in the order of the chromosomal location of their encoding genes are shown on the top. DNA copy number of chromosomes XII and XVI are shown in red. Arrows indicate an amplification of part of chromosome III (red) and a loss of part of chromosome XV (green). The graph below shows the average DNA copy number per chromosome relative to strain YJM308. The graph below shows RNA copy number averaged per chromosome. Error bars represent the SD of the chromosome means from two biological replicates. Medians are identical to the means. Asterisk indicate significant deviations from the expected value as determined by a one sample *t*-test (p < 0.01). (**E**) DNA and RNA copy number analysis of strain Y2189 compared to Y2209. Data are represented as in (**D**). Error bars represent the SD of the chromosome means from two biological replicates. Medians are identical to the means. Asterisk indicate significant deviations from the expected value as

*Figure 1 continued on next page*

Figure 4A), and WE372, which is trisomic for chromosome I to harbor stable karyotypes (*Figure 2A*). However, DNA copy numbers in strains UC5, Y3, Y6, and CBS7960 exhibited non-integer DNA copy number states indicating that the strains are heterogeneous (*Figure 2B–E*).

Analysis of the karyotypes of the other 35 wild strains (both Figure 1 and Supplementary file 1 in *Hose et al. (2015)*) revealed that more than half of the strains harbored karyotype profiles consistent with heterogeneity. Importantly, strains K10, YJM308, and Y2209 utilized as the euploid reference in the gene expression analysis of the aneuploid wild strains YJM428, Y2189, K1, and K9 (Figure 2 in *Hose et al., 2015*) appeared to harbor heterogeneous karyotypes (*Figure 2F–H*). In particular, strain YJM308 harbors an amplification of chromosome XV and has lost part of chromosome III (*Figure 2G*). We conclude that only 10.6% (5 out of 47) of the strains analyzed by *Hose et al. (2015)* harbor relatively stable aneuploidies that are confined to 1 – 2 chromosomes.

As all strains studied by *Hose et al. (2015)* were derived from single colonies, our finding of significant karyotype heterogeneity indicates that a large fraction of wild yeast strains grown under standard laboratory conditions are unstable. The observed instability and heterogeneity of many wild *S. cerevisiae* strains makes it likely that the aneuploidies in these wild isolates are a consequence of culturing the natural variants under laboratory conditions to which they may be ill-adapted to, instead of these strains being naturally aneuploid. Caution is therefore warranted when analyzing growth rates, gene expression patterns and phenotypes of such wild yeast strains under laboratory growth conditions.

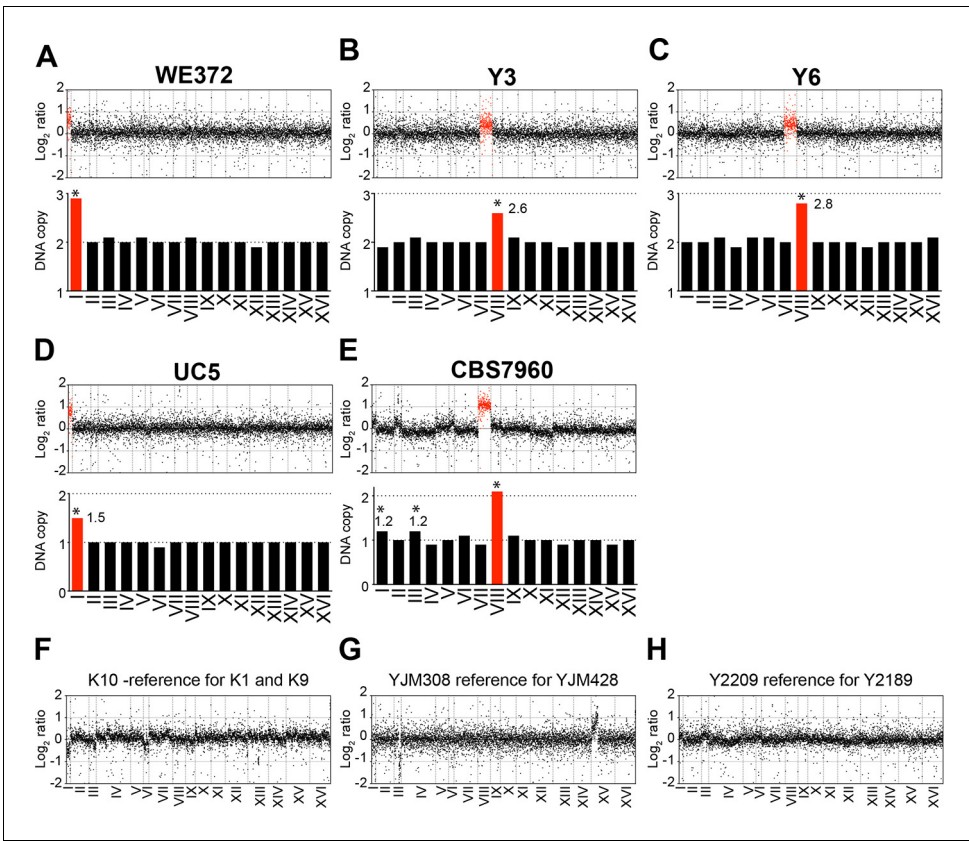

**Figure 2.** Karyotypes of aneuploid wild *S. cerevisiae* strains Y3, Y6 UC5, CBS7960, and WE372 and euploid control strains. (A–E) Relative DNA copy of WE372 (**A**), Y3 (**B**), Y6 (**C**), UC5 (**D**), and CBS7960 (**E**) compared to S288C. Log$_2$ (aneuploid vs. euploid DNA) per gene relative (top) are shown in the order of the chromosomal location of their encoding genes. DNA copy numbers of amplified chromosomes are shown in red. Bar graphs (bottom) represent the DNA copy numbers averaged per chromosome. Asterisks indicate significant deviations from expected integral value using one sample t test (p < 0.01). (F–G) Relative DNA copy of K10 (**F**), YJM308 (**G**), and Y2189 (**H**) compared to S288C. Log$_2$ ratio (aneuploid vs. euploid DNA) per gene are shown in the order of the chromosomal location of their encoding genes.

## Gene expression levels correlate with gene copy number in wild aneuploid *S. cerevisiae* strains

In our previous studies, we found RNA and DNA levels to be well-correlated in haploid laboratory W303 strains carrying additional chromosomes (*Dephoure et al., 2014*; *Torres et al., 2007*). *Hose et al. (2015)* reported that this coordination between DNA and RNA levels was not evident in wild budding yeast isolates. Their conclusion was based on three analyses. In the first analysis, they characterized six wild aneuploid isolates; in the second, they studied three euploid-aneuploid strain pairs; and in the third analysis, they investigated two sets of strains each comprised of a series of strains with increasing aneuploidies of one particular chromosome. To begin to understand the mechanisms that could have led to the loss of dosage compensation mechanisms in laboratory strains, we reanalyzed the data generated by *Hose et al. (2015)* using the methods we previously employed to examine the effects of aneuploidy on gene expression in laboratory strains.

## Analysis of wild yeast strains YJM428, Y2189, YPS1009, NCYC110, K1, and K9

*Hose et al. (2015)* compared mRNA levels with DNA copy number of amplified genes across six aneuploid wild yeast strains called K9, YPS1009, NCYC110, YJM428, Y2189, and K1 and concluded that 38% (838 of 2,204) of amplified genes showed lower expression than predicted by their gene copy number (light blue points in Figure 4A in *Hose et al., 2015*). We reevaluated their findings. Because of karyotype heterogeneity in strains YJM428, Y2189, and K1 (*Figure 1D–F*), we did not reanalyze these strains except to determine the false discovery rate discussed in detail below.

Strains YPS1009 and K9 are trisomic for chromosomes XII and IX+X, respectively, while NCYC110 harbors a tetrasomy of chromosome VIII. Our analysis revealed that the expression of all genes on the aneuploid chromosomes increased proportionally with gene copy number (*Figure 1A–C*, *3A, B*). As predicted by a null model with no compensation, we found that the $\log_2$ ratios of expression values of genes encoded by the triplicated chromosomes of these strains to fit a normal distribution with a mean value very close to the predicted $\log_2$ ratio of 0.58 (mean $\log_2$ ratio = 0.55, $R^2$ = 0.99, *Figure 3A*, middle panel, *Figure 3B*) for the trisomic strains and a $\log_2$ ratio of 1 (mean $\log_2$ ratio = 0.95, $R^2$ = 0.97, *Figure 3A*, right panel) for the tetrasomic chromosome. No skewness in the distributions - more compensating or exacerbating - was noted as would be expected if a large fraction of the genes encoded on the aneuploid chromosome were dosage compensated (skewness = 0.02 (3n) and 0.07 (4n); *Figure 3A*). The distribution of $\log_2$ ratios of expression values of genes encoded by euploid chromosomes also fit a normal distribution with the predicted $\log_2$ ratio of 0 (mean $\log_2$ ratio = 0.00, $R^2$ = 0.99, *Figure 3A* left panel). These data are very much in line with what is observed in aneuploid laboratory strains. RNA quantification of two disomic W303 strains (disomes V and XVI) showed that the $\log_2$ ratios of expression values of genes encoded by the duplicated chromosomes fit a normal distribution with a mean value very close to the predicted $\log_2$ ratio of 1 (mean $\log_2$ ratio = 1.03, $R^2$ = 0.98, *Figure 3C*).

To determine how many genes were potentially subject to dosage compensation, we used 2 SD from the means of the $\log_2$ ratios of each amplified chromosome and found that between 0% (0 gene in NCYC110) and 3% (19 genes in K9) of amplified genes showed values lower than expected (*Table 1*). Importantly, a similar number of genes was found to exhibit higher than expected expression (between 1% in YPS1009 and 2% in NCYC110, *Table 1*). Using the same cutoff on the euploid chromosomes, we found between 0.1% (7 genes in NCYC110) and 3% (153 genes in K1) genes with values lower than expected. The nature of the distributions of gene expression patterns (normal distribution with expected means) and these analyses are inconsistent with high levels of dosage compensation occurring in wild yeast strains. Instead, they indicate that gene expression proportionally increases with copy number without signs of dosage compensation in wild yeast strains. The fact that the euploid chromosomes encode the same proportion of up and downregulated genes as the aneuploid chromosomes further indicates that any effects on gene expression seen in these strains are likely to be the consequence of measurement noise or a transcriptional response elicited by the aneuploid state rather than dosage compensation.

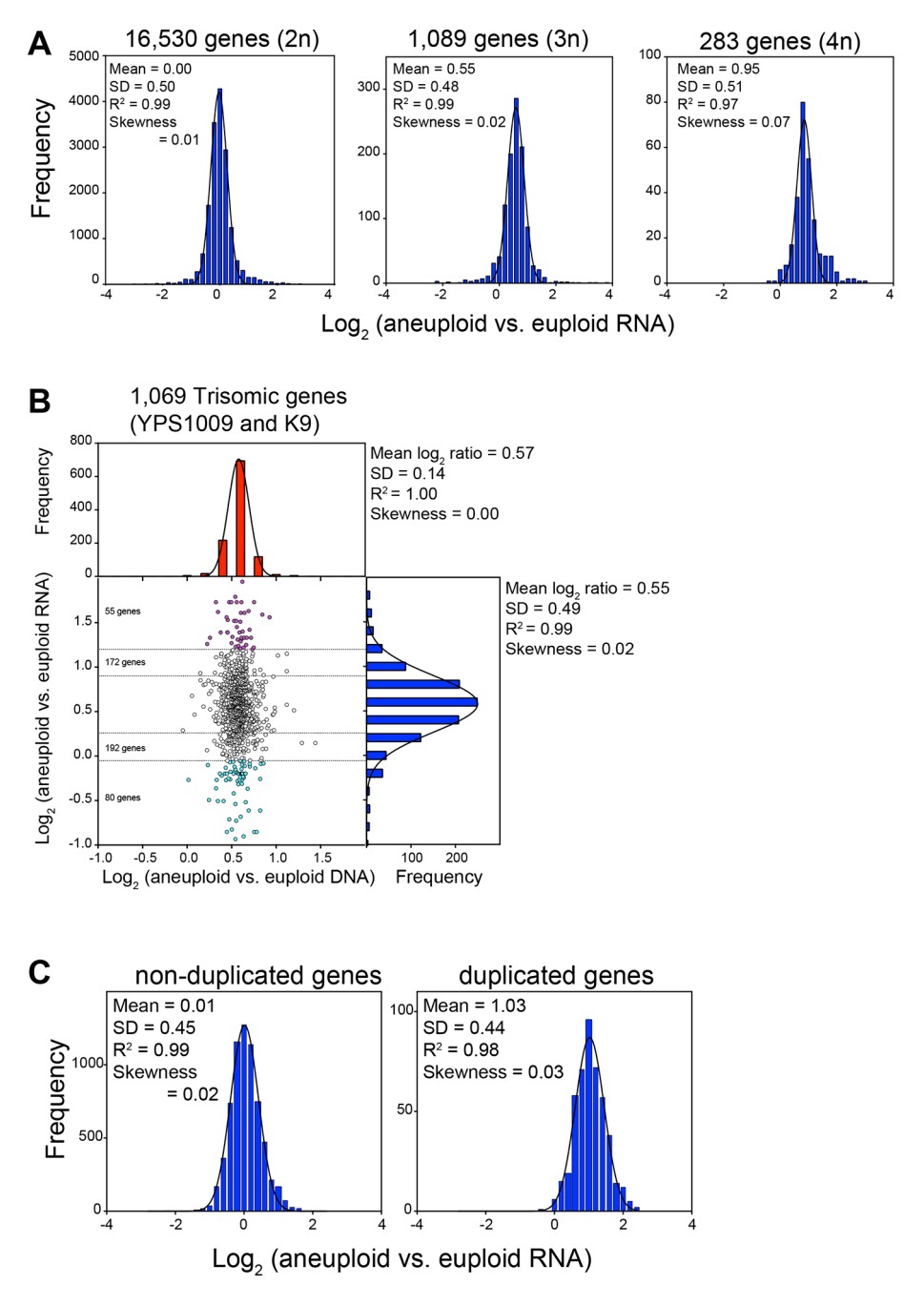

**Figure 3.** RNA levels correlate with DNA copy number in wild and laboratory strains of *S. cerevisiae*. (**A**) Histogram of the log₂ ratios of the RNA copy number of genes located on euploid chromosomes (left panel, strains YPS1009, NCYC110, and K9), genes present on trisomic chromosomes (3n, middle panel, YPS1009, and K9), and genes present on tetrasomic chromosomes (right panel, NCYC110), relative to euploid controls are shown. Bin size for all histograms is log₂ ratio of 0.2, medians are identical to means. Fits to a normal distribution (black line), means and goodness of fit ($R^2$) and skewness are shown for each distribution. (**B**) The average log₂ (aneuploid vs. euploid RNA) of triplicated genes plotted against average log₂ (aneuploid vs. euploid DNA) in strains YPS1009 and K9. Histogram of the log₂ ratios of the DNA copy number is shown in red (mean log₂ ratio = 0.57, SD = 0.14, $R^2$ = 1.0, skewness = 0.00). Histogram of the log₂ ratios of the RNA copy number of is shown in blue (median = mean = 0.55, skewness = 0.02). Fits to a normal distribution are shown (black line). Numbers of genes that show RNA copy numbers lower or higher than 1 or 2 SD from the mean are shown (separated by dotted lines). (**C**) Histogram of the log₂ ratios of the RNA copy number of genes located on euploid chromosomes (left panel), and

*Figure 3 continued on next page*

## Analysis of the aneuploid strain pairs YPS163, T73, and YJM428

To further characterize dosage compensation in wild variants *Hose et al. (2015)* generated a panel of isogenic euploid and aneuploid strain pairs. They isolated a disomic strain for chromosome VIII (YPS163-chr VIII-2n) of the euploid strain YPS163, and euploid versions of strain T73, which is tetrasomic for chromosome VIII (T73-chrVIII-4n) and of strain YJM428, which is tetrasomic for chromosome XVI (YJM428-chrXVI-4n). They then determined DNA copy number state and gene expression levels in these strains and concluded that between 11 and 36% of genes were expressed at lower than expected levels, that is, they were dosage compensated.

We compared the average chromosome copy number in the three aneuploid strains with the average RNA copy number in these strains and found that RNA levels proportionally increased with DNA copy number (*Figure 4A,B*). The aneuploidies in the three strains represent duplications. We were, therefore, able to combine the duplicated values of the DNA and RNA copy of all the three strains. The 941 duplicated genes showed a mean $\log_2$ ratio of 1.02 (SD = 0.29, $R^2$ = 0.99) for DNA copy number and a nearly identical mean $\log_2$ ratio (mean = 0.97; SD = 0.36, $R^2$ = 0.99) for RNA copy number (*Figure 4C*). Furthermore, the distribution of expression values fit a normal distribution and was indistinguishable from the distribution of the gene expression values of genes encoded by the euploid chromosomes. The standard deviations of the RNA distributions were similar for euploid and aneuploid chromosomes (*Figure 4C* bottom graphs) and each distribution showed skewness of 0.01 and 0.02, respectively. These observations indicate that the variance of the euploid genes is the same as that of the aneuploid genes. If dosage compensation were to occur, variance and skewness, would be different between genes encoded by euploid and aneuploid chromosomes. Lastly, using 2 SD as cutoff to find potential dosage compensated genes, we identified a small number of outliers. Importantly, the number of up and downregulated outliers was similar (*Figure 5*). We conclude that RNA levels correlate well with DNA copy number in aneuploid strains YPS163, T73, and YJM428.

## Analysis of the aneuploid strain series YPS1009 and NCYC110

The third set of strains that *Hose et al. (2015)* analyzed was comprised of two series of yeast strains that carry increasing numbers of a specific chromosome. Starting with strain YPS1009, which carries three copies of chromosome XII (YPS1090-chrXII-3n), *Hose et al. (2015)* derived a euploid strain (YPS1009-chrXII-2n) and a strain that is tetrasomic for chromosome XII (YPS1009-chr XII-4n; *Figure 6A,B*). Using strain NCYC110, which carries four copies of chromosome VIII (NCYC110-chrVIII-4n), they isolated a strain trisomic for chromosome VIII (NCYC110-chrVIII-3n) and a diploid strain (NCYC110-chrVIII-2n; *Figure 6A,B*). They then determined DNA copy number state and gene expression levels in these strain series and concluded that 11% of genes encoded on chromosome VIII and 29% of genes encoded on chromosome XII were dosage compensated.

We found the gene expression distribution of genes located on euploid and aneuploid chromosomes to fit normal distributions without any skewness (*Figure 6C,D*). The two trisomic strains YPS1009-chrXII-3n and NCYC110-chrVIII-3n together harbored 776 triplicated genes. Their averaged $\log_2$ ratio of DNA copy number was 0.57 (SD = 0.15, $R^2$ = 0.99) and 0.60 (SD = 0.53, $R^2$ = 0.97) for RNA copy number (*Figure 6C*). A similar coordination between DNA and RNA copy number was observed in the tetrasomic strains. The mean $\log_2$ ratio of DNA copy number of genes located on the tetrasomic chromosome was 0.99 (SD = 0.16, $R^2$ = 0.99), the mean mRNA expression of genes located on the tetrasomic chromosome was 0.99 (SD = 0.62, $R^2$ = 0.96; *Figure 6D*). Importantly, the distributions of DNA and mRNA copy number were similar for genes located on euploid, trisomic and tetrasomic chromosomes, with similar SDs and no evidence of skewness (skewness varied between 0.00 and 0.03).

In summary, we were not been able to detect dosage compensation in the strains described in *Hose et al. (2015)*. RNA levels of genes encoded by the aneuploid chromosomes are normally distributed with the expected or close to expected mean. No difference was observed between the number of down-regulated genes located on aneuploid and euploid chromosomes. Furthermore, no skewness was observed for any of the distributions. *Figure 7* shows how distributions exhibit negative values of skewness when dosage compensation occurs. Importantly, in a previous study, we were able to detect attenuation in the expression of certain genes in aneuploid yeast strains using the method employed here. In *Dephoure et al. (2014)*, we examined the proteomes of haploid disomic laboratory yeast strains and found that production of ribosomal proteins encoded on

**Table 1.** DNA and RNA copy number of six wild *S. cerevisiae* strains. The columns describe the following parameters: Column 1: Strain name. Column 2: Identity of chromosomes amplified in each strain. Euploid represents the combined data of all euploid chromosomes in a given strain. Column 3: Reported chromosome copy number. Column 4: Number of genes quantified by RNA-seq. Column 5: Mean of the normalized $\log_2$ ratios (aneuploid vs. euploid RNA). Column 6: Standard deviation (SD) of the normalized $\log_2$ ratios (aneuploid vs. euploid RNA). Column 7: Mean of the normalized $\log_2$ ratios (aneuploid vs. euploid DNA). Column 8: Standard deviation (SD) of the normalized $\log_2$ ratios (aneuploid vs. euploid DNA). Column 9: Number of genes whose values are below two SD from the mean. Column 10: Number of genes whose values are above two SD from the mean. Column 11: Cutoff used by *Hose et al. (2015)* to identified genes that are dosage compensated.

| 1 | 2 | 3 | 4 | 5 | 6 | 7 | 8 | 9 | 10 | 11 |
|---|---|---|---|---|---|---|---|---|---|---|
| STRAIN | Chr | Copy number | Genes | RNA Mean | RNA SD | DNA Mean | DNA SD | Number of genes RNA <2*SD | Number of genes RNA >2*SD | Cutoffs by Hose et al |
| YJM428 -1 | XII | 3 | 525 | 0.52 | 0.63 | | | | 15 | |
| | XVI | 4 | 485 | 0.95 | 0.66 | | | 9 | 17 | |
| | Euploid | 2 | 5087 | −0.01 | 0.72 | | | 116 | 169 | |
| YJM428-2 | XII | 3 | 533 | 0.54 | 0.70 | 0.60 | 0.22 | 10 | 18 | N/A |
| | XVI | 4 | 490 | 0.92 | 0.63 | 0.96 | 0.23 | 11 | 18 | N/A |
| | Euploid | 2 | 5160 | −0.01 | 0.72 | 0.00 | 0.28 | 75 | 183 | |
| | | | | | | Aneuploid genes | | 9 (1%) | 14 (1%) | |
| | | | | | | Euploid genes | | 36 (1 %) | 77 (1%) | |
| Y2189-1 | I | 4 | 88 | 0.77 | 0.89 | | | 3 | 4 | |
| | III | 3 | 170 | 0.60 | 0.88 | | | 5 | 3 | |
| | IX | 3 | 216 | 0.42 | 0.91 | | | 5 | 9 | |
| | XI | 3 | 325 | 0.37 | 0.89 | | | 3 | 8 | |
| | Euploid | | 5209 | 0.05 | 0.76 | | | 104 | 204 | |
| Y2189-2 | I | 4 | 89 | 0.63 | 1.01 | 1.05 | 1.04 | 4 | 3 | 0.21 |
| | III | 3 | 167 | 0.53 | 0.90 | 0.53 | 0.55 | 5 | 6 | 0.24 |
| | IX | 3 | 214 | 0.37 | 0.96 | 0.45 | 0.48 | 5 | 9 | N/A |
| | XI | 3 | 324 | 0.46 | 0.65 | 0.47 | 0.25 | 3 | 10 | 0.13 |
| | Euploid | 2 | 5231 | 0.06 | 0.77 | 0.00 | 0.43 | 142 | 165 | |
| | | | | | | Aneuploid genes | | 9 (1%) | 15 (2%) | |
| | | | | | | Euploid genes | | 50 (1%) | 124 (2%) | |
| YPS1009-1 | XII | 3 | 511 | 0.53 | 0.62 | | | 13 | 27 | |
| | Euploid | 2 | 5482 | 0.00 | 0.57 | | | 132 | 136 | |
| YPS1009-2 | XII | 3 | 520 | 0.49 | 0.73 | | | 16 | 20 | |
| | Euploid | 2 | 5531 | 0.00 | 0.60 | | | 145 | 119 | |
| YPS1009-3 | XII | 3 | 521 | 0.56 | 0.66 | 0.62 | 0.24 | 11 | 31 | 0.10 |
| | Euploid | 2 | 5532 | 0.00 | 0.56 | 0.00 | 0.31 | 130 | 180 | |
| | | | | | | Aneuploid genes | | 5 (1%) | 5 (1%) | |
| | | | | | | Euploid genes | | 46 (1%) | 27 (0%) | |
| NCYC110-1 | VIII | 4 | 288 | 0.97 | 0.61 | | | 3 | 14 | |
| | Euploid | 2 | 5806 | 0.00 | 0.62 | | | 69 | 274 | |
| NCYC110-2 | VIII | 4 | 294 | 0.93 | 0.59 | | | 4 | 14 | |
| | Euploid | 2 | 5919 | 0.00 | 0.61 | | | 60 | 247 | |
| NCYC110-3 | VIII | 4 | 292 | 0.98 | 0.58 | 0.98 | 0.16 | 4 | 14 | 0.10 |
| | Euploid | 2 | 5890 | 0.00 | 0.57 | 0.00 | 0.12 | 61 | 254 | |
| | | | | | | Aneuploid genes | | 0 (0%) | 5 (2%) | |
| | | | | | | Euploid genes | | 7 (0%) | 102 (2%) | |

*Table 1 continued on next page*

disomic chromosomes is significantly attenuated causing the distributions to exhibit negative skewness (*Dephoure et al., 2014*).

## Evaluation of the analysis methods employed by *Hose et al. (2015)*

Why did *Hose et al. (2015)* arrive at such different conclusions than we did? To address this question, it is important to understand how *Hose et al. (2015)* analyzed and interpreted their data.

We identified two problems in their data analysis. The first regards data normalization. The ratios are off by a factor of $\log_2 = 0.1$–$0.2$ (normalized data utilized in *Hose et al. (2015)* were kindly provided by A. Gasch). Most normalization protocols do not take into account that aneuploid strains harbor a different number of gene copies compared to euploid strains. When this is not manually corrected, data are shifted by a factor that depends on the degree of aneuploidy and results in incorrect values as shown in *Figure 8A*. The degree by which the data used for analysis by *Hose et al. (2015)* deviate from the correctly normalized expression values is of the same magnitude as some of the cutoffs used to define dosage compensated genes (detailed next).

The second problem with the data analysis concerns cutoffs used to define dosage compensated genes. To establish cutoffs for designating whether a gene is dosage compensated or not *Hose et al. (2015)* used the SD of the DNA measurements, which ranged between 0.1 and 0.45 (*Table 1* column 11, data kindly provided by A. Gasch) as cutoffs for the RNA measurements (Figure 4 in *Hose et al., 2015*). Genes whose expression deviated by the DNA SD value from the expected RNA expression level were considered dosage compensated. This is not the correct cutoff tool because the DNA copy number measurements are less variable than mRNA measurements. As seen in *Figure 8B*, transcript levels can vary by several orders of magnitude depending on the expression levels of a particular gene. Therefore, the distributions of relative RNA changes will show bigger SDs than gene copy number distributions. Indeed, the RNA measurements conducted by *Hose et al. (2015)* show SDs between 0.53 and 1.01 (*Table 1*, column 6). Employing the SD derived from the DNA measurements, which are fairly lower (*Table 1*, column 8), will therefore not identify genes that are dosage compensated in a statistically significant manner (see false discovery rate discussion below). This is of particular importance as genes identified in Figure 4 of *Hose et al. (2015)* as dosage compensated were included in a group of 245 dosage compensated genes used to establish GO term enrichments among dosage compensated genes.

To determine how *Hose et al. (2015)* identified 838 of 2204 genes as dosage compensated we re-evaluated their analysis. Figure 4A in *Hose et al. (2015)* displays an unusual behavior. The null model shown by the diagonal of equal RNA and DNA in this figure did not bisect the blue (compensated) and magenta (exacerbated) points. Instead, the vast majority of points below this line were considered compensated while the vast majority of points above this line were considered not exacerbated. This suggests that there could be a high number of false positives amongst the 838 genes determined to be dosage compensated.

To address this possibility, we used two methods to determine the false discovery rate. First, we scrambled the data by randomly permuting the RNA/DNA ratio between genes. We did this independently for each replicate. This preserves the RNA/DNA ratios but unlinks the values from their replicate measurements and genes. Then, we used the same effective significance cutoffs used by *Hose et al. (2015)* to determine the number of dosage compensated genes (see Materials and methods). As this is a randomized dataset, genes identified by this method are noise and can be used to determine the number of genes the analysis method would find just by chance. Based on 10,000 randomizations, we determined that on average, 779 genes would have passed the threshold method used by *Hose et al. (2015)* by chance. This yields a false discovery rate (FDR) of 92.9%. This high false discovery rate was also seen at much lower cutoffs. The FDR was between 93 and 100% at cutoffs from 0.1 STD to 2 STDs.

Second, we calculated the average SD for each RNA sequencing measurement. As the DNA measurements for each strain were not reported independently, we calculated the average chromosome-wide DNA error from all the sequencing data that were deposited and used the lowest of these as an estimate for all analyses. We combined these errors together (square root of summed squares of the two composite noises) to give a measurement noise distribution for the experiment. We then randomly sampled from a normal distribution where the SD for this normal distribution was randomly sampled from the measurement noise distribution. Using this method, we found that on average, 754 genes would have passed the effective threshold used by *Hose et al. (2015)*. This

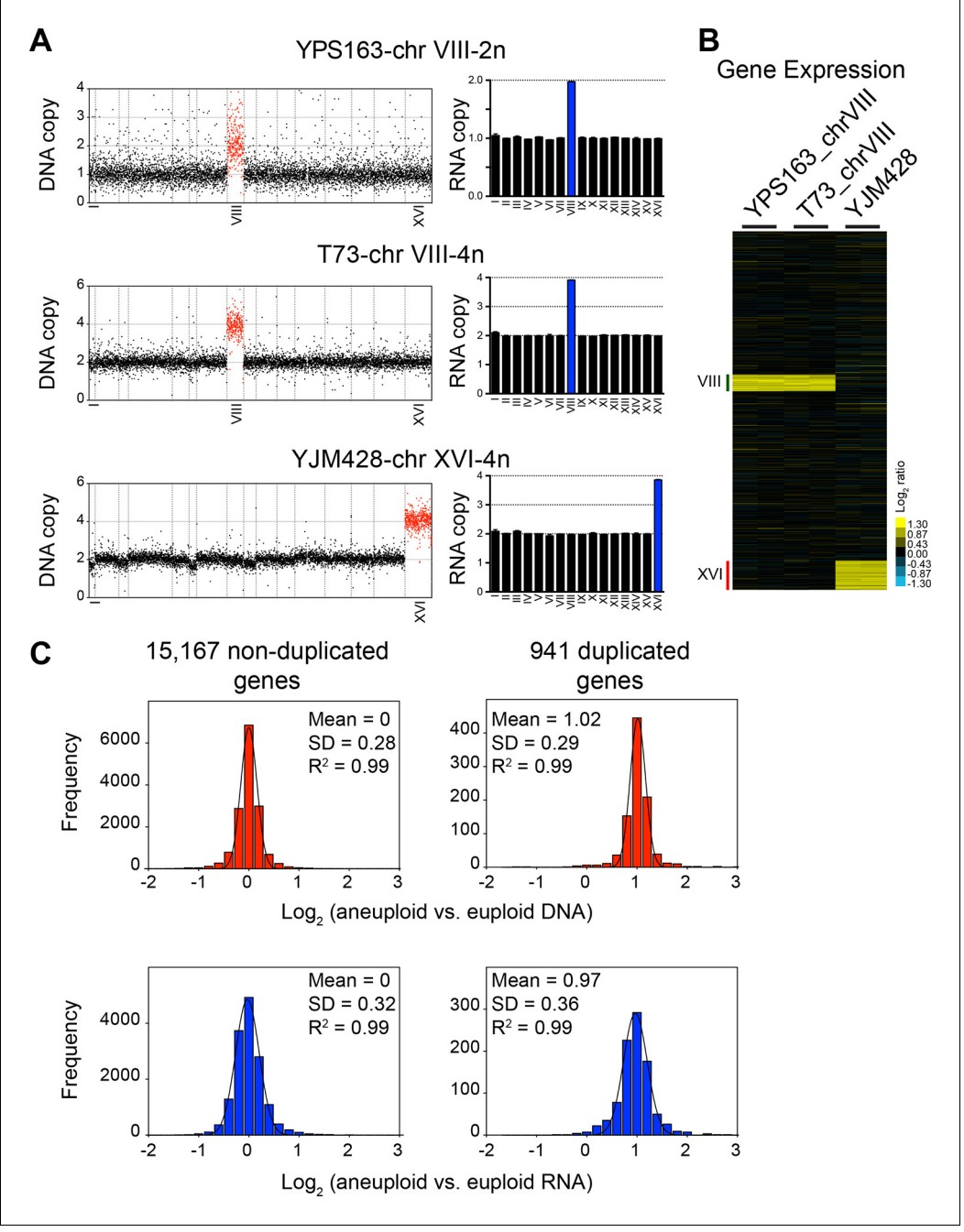

**Figure 4.** DNA and RNA copy number of euploid and aneuploid isogenic wild *S. cerevisiae* strains. (**A**) Plots for strains YPS163-chrVIII-2n, T73-chrVIII-4n, and YJM428-chrXVI-4n, represent the $\log_2$ ratio of their relative DNA copy number compared to their isogenic and euploid counterparts. DNA copy numbers are shown in the order of the chromosomal location of their encoding genes (left). DNA copy numbers of amplified chromosomes are shown in red. Bar graphs on the right represent the RNA copy numbers averaged per chromosome for aneuploid strains relative to euploid reference strains. The average RNA copies of non-amplified chromosomes are shown in black. Amplified chromosomes, as predicted by the karyotype, are shown in blue. (**B**) Gene expression of three aneuploid strains ordered by chromosome position. Experiments (columns) of two biological replicates are shown. (**C**) Histogram of the $\log_2$ ratios of the DNA (top) and RNA (bottom) copy number of genes located on euploid chromosomes (left) and genes located on duplicated chromosomes (right) relative to euploid controls are shown. Bin size for all histograms is $\log_2$ ratio of 0.2, medians are identical to means and all distributions show a skewness of 0.01. Fits to a normal distribution are shown (black line) and so are means and goodness of fit ($R^2$) for each distribution.

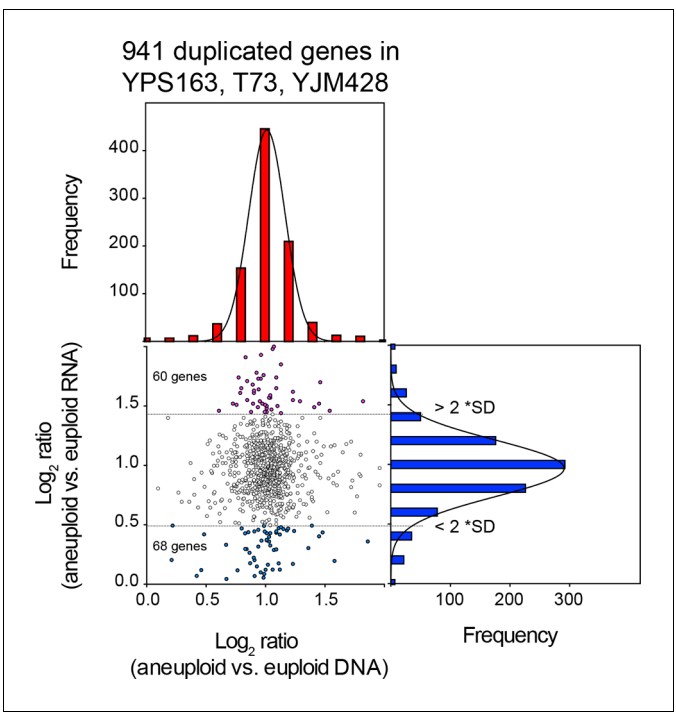

**Figure 5.** Comparison of DNA and RNA copy number distributions of strains YPS163, T73, and YJM428. The average $\log_2$ (aneuploid vs. euploid RNA) of 941 genes located on duplicated chromosomes plotted against the average $\log_2$ (aneuploid vs. euploid DNA) in strains YPS163, T73, and YJM428. Histogram of the $\log_2$ ratios of the DNA copy number is shown in red. Histogram of the $\log_2$ ratios of the RNA copy number is shown in blue. Fits to a normal distribution are shown (black line). The number of genes that show RNA copy numbers lower or higher than 2 SD from the mean are shown (separated by dotted lines).

corresponds to a false discovery rate of 89%; this value is likely a small underestimation of FDR given our method for estimation of DNA error. We conclude that both methods that we applied to determine false discovery rate strongly suggest that only a handful, at most ~70 genes or <3%, are actually dosage compensated. These results are completely in line with previous findings from laboratory strains (*Springer et al., 2010*; *Dephoure et al., 2014*; *Torres et al., 2007*).

In a second approach to identify dosage compensated genes, *Hose et al. (2015)* defined genes to be dosage compensated when the RNA levels did not increase with DNA copy number in their YPS1009 (2N, 2N+1 chromosome XII, 2N+2 chromosomes XII) and NCYC110 (2N, 2N+1 chromosome VIII, 2N+2 chromosomes VIII) ploidy series. For this, they developed a mixture of linear regression (MLR) model to classify genes based on the slope and intercept of the RNA-gene copy relationships. When RNA levels did not increase proportionately as DNA copy increased as evidenced by slopes lower than 1 in the MLR model, a gene was classified as dosage compensated and categorizes as Class 3a in Table 1 in *Hose et al. (2015)*. Thirty genes on chromosome VIII and 142 genes on chromosome XII were identified as dosage compensated through this method. This method of identifying dosage compensated genes is problematic in several ways. First, because there are only three data points per analysis, a single deviating data point can have a significant impact on the slope. For example, a gene with values of $\log_2$ ratio = 0.3, 0.6 and 0.8 representing, two, three, and four copies, respectively, will perfectly fit a straight line with the slope of 0.5 and hence would be classified as dosage compensated according to the criteria in *Hose et al. (2015)* even though none of the three data points significantly deviates from the mean value given a SD of 0.3 or higher (*Figure 8C*, *Table 2*). In fact, the majority (103 of 172) of class 3a genes (Table 1 and Supplemental File 3 in *Hose et al. (2015)*) fit the MLR model with slopes of 0.5 or higher indicating that their gene expression increases with copy number.

Because of these considerations, we reanalyzed the dosage compensation in chromosomes VIII and XII by two methods. In the first, we calculated the mean and standard deviations for each of the

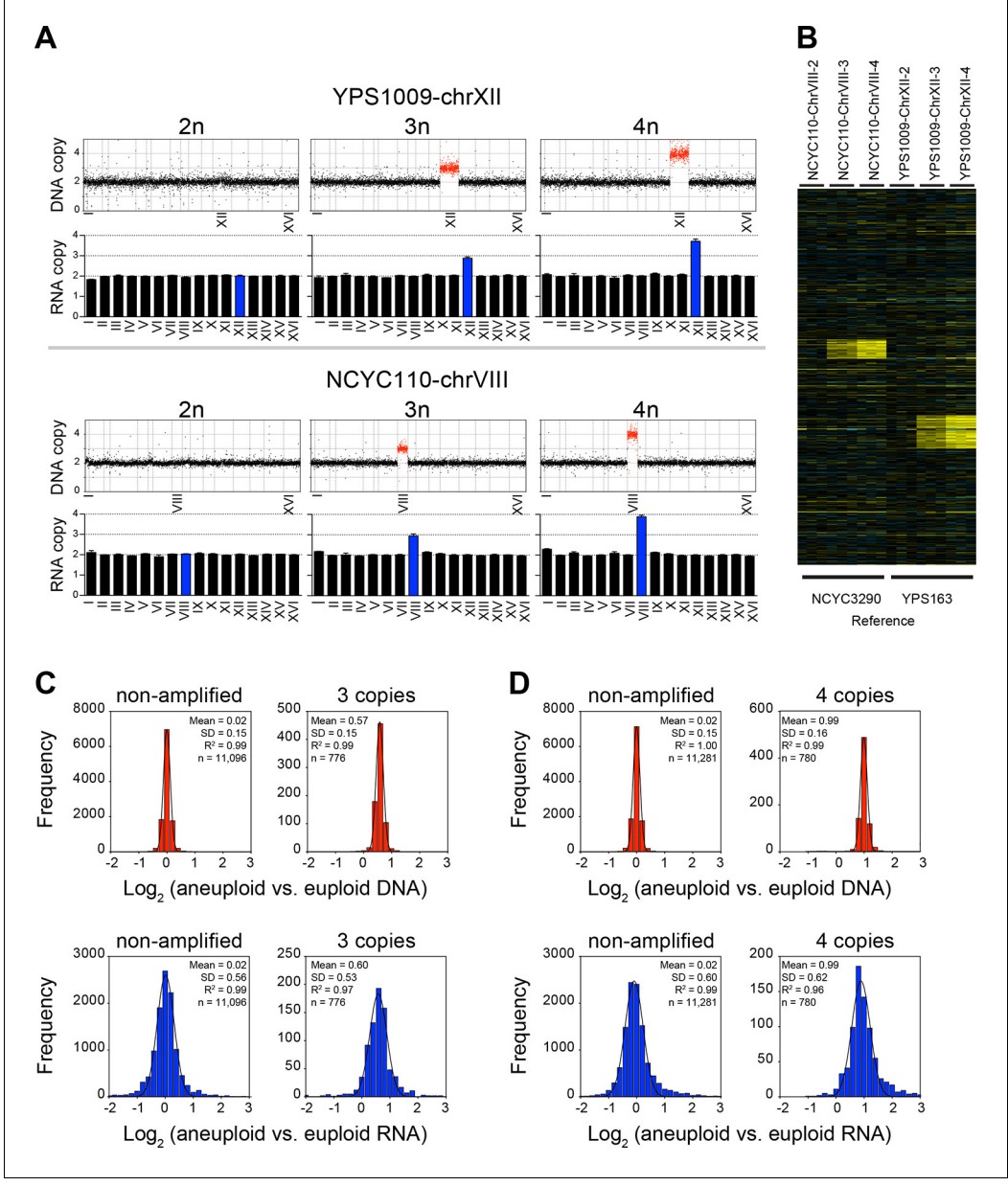

**Figure 6.** RNA copy number proportionally increases with DNA copy number in aneuploid series of wild *S. cerevisiae* strains. (A) Plots for strain series YPS1009-XII-2n, YPS1009-XII-3n, YPS1009-XII-4n and strain series NCYC110-chrVIII-2n, NCYC110-chrVIII-3n, NCYC110-chrVIII-4n represent the DNA copy number compared to their euploid counterparts. DNA copy numbers are shown in the order of the chromosomal location of their encoding genes. DNA copy numbers of amplified chromosomes are shown in red. Bar graphs below represent the RNA copy numbers averaged per chromosome for aneuploid strains relative to euploid reference strains. The average RNA copies of non-amplified chromosomes are shown in black. Amplified chromosomes, as predicted by the karyotype, are shown in blue. (B) Gene expression of strain series YPS1009-XII-2n, YPS1009-XII-3n, YPS1009-XII-4n, and strain series NCYC110-chrVIII-2n, NCYC110-chrVIII-3n, NCYC110-chrVIII ordered by chromosome position. Experiments (columns) of three biological replicates are shown. (C) Histogram of the $\log_2$ ratios of the DNA copy number of genes located on euploid chromosomes (top left) and genes located on trisomic chromosomes (top right) in strains YPS1009-chrXII-3n and NCYC110-chrVIII-3n relative to euploid controls are shown. Fits to a normal distribution are shown (black line). Histogram of the $\log_2$ ratios of the RNA copy number of genes located on euploid chromosomes (bottom left) and genes present on trisomic chromosomes (bottom right) in strains YPS1009-chrXII-3n and NCYC110-chrVIII-3n relative to euploid controls are shown. Fits to a normal distribution are shown (black line). (D) Histogram of the $\log_2$ ratios of the DNA copy number of genes located on euploid chromosomes (top left) and genes located on tetrasomic chromosomes (top right) in strains YPS1009-chrXII-4n and NCYC110-chrVIII-4n relative to euploid controls are shown. Fits to a normal distribution are shown (black line). Histogram of the $\log_2$ ratios of the RNA copy number of genes located on euploid chromosomes (bottom left) and genes located on tetrasomic chromosomes (bottom right) in strains YPS1009-chrXII-4n and NCYC110-chrVIII-4n relative to euploid controls are shown. Fits to a normal distribution are shown (black line).

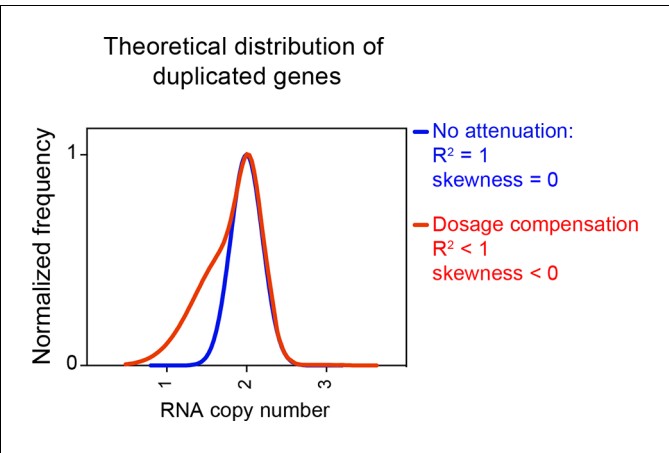

**Figure 7.** Theoretical distribution of RNA copy number of dosage compensated duplicated genes. The theoretical distribution of RNA copy number of duplicated genes when no dosage compensation takes place is shown in blue. The theoretical distribution of RNA copy number of duplicated genes when 30% of the genes are dosage compensated is shown in red. The fit to a normal distribution shows negative skewness values (red).

biological replicates in the NCYC110 and YPS1009 strain series and found that only two genes on aneuploid chromosome VIII and seven genes on aneuploid chromosome XII show $\log_2$ ratios 1 SD lower than the mean in three biological replicates and were reproducibly lower when present in 3 or 4 copies. Not a single gene passed the cutoff of 2 SD below the mean. We conclude that for the majority of genes only one of the two data points supports the conclusion that a gene is expressed at lower than the expected value, calling into question that the genes identified by this approach are indeed dosage compensated.

In a second approach, we defined the false discovery rate (not determined by *Hose et al., 2015*) to determine whether the genes identified as dosage compensated were statistically distinguishable from noise. Using the same subset of genes that *Hose et al. (2015)* examined, we calculated a slope based on the nine RNA measurements and matching DNA measurements (three replicates of three strains) for both YPS1009 and NCYC110. Using the genes identified as dosage compensated by *Hose et al. (2015)*, we determined the effective cutoff of their MLR method (see Materials and methods). We then randomly permuted the positions of the RNA and DNA data and recalculated the slopes for each gene. From this analysis we determined the false discovery rate was within error of 100%. We conclude that there is no significant dosage compensation in these aneuploid series.

## Discussion

Our analyses indicate that a large fraction of wild *S. cerevisiae* strains are unstable and heterogeneous when grown under laboratory conditions. This result suggests that at least some wild *S. cerevisiae* strains may not be naturally aneuploid but could become aneuploid due to an adaptive response to laboratory growth conditions. Reevaluation of the DNA and RNA copy number data generated by *Hose et al. (2015)* further indicates that dosage compensation is rare in both wild and laboratory strain of *S. cerevisiae*. Both types of strains lack mechanisms that allow them to attenuate gene expression in response to gene copy number alterations. We conclude that wild variants of *S. cerevisiae* do not have mechanisms in place that protect them from changes in gene copy number. Their regulation of gene expression is thus the same as that of laboratory strains of budding yeast.

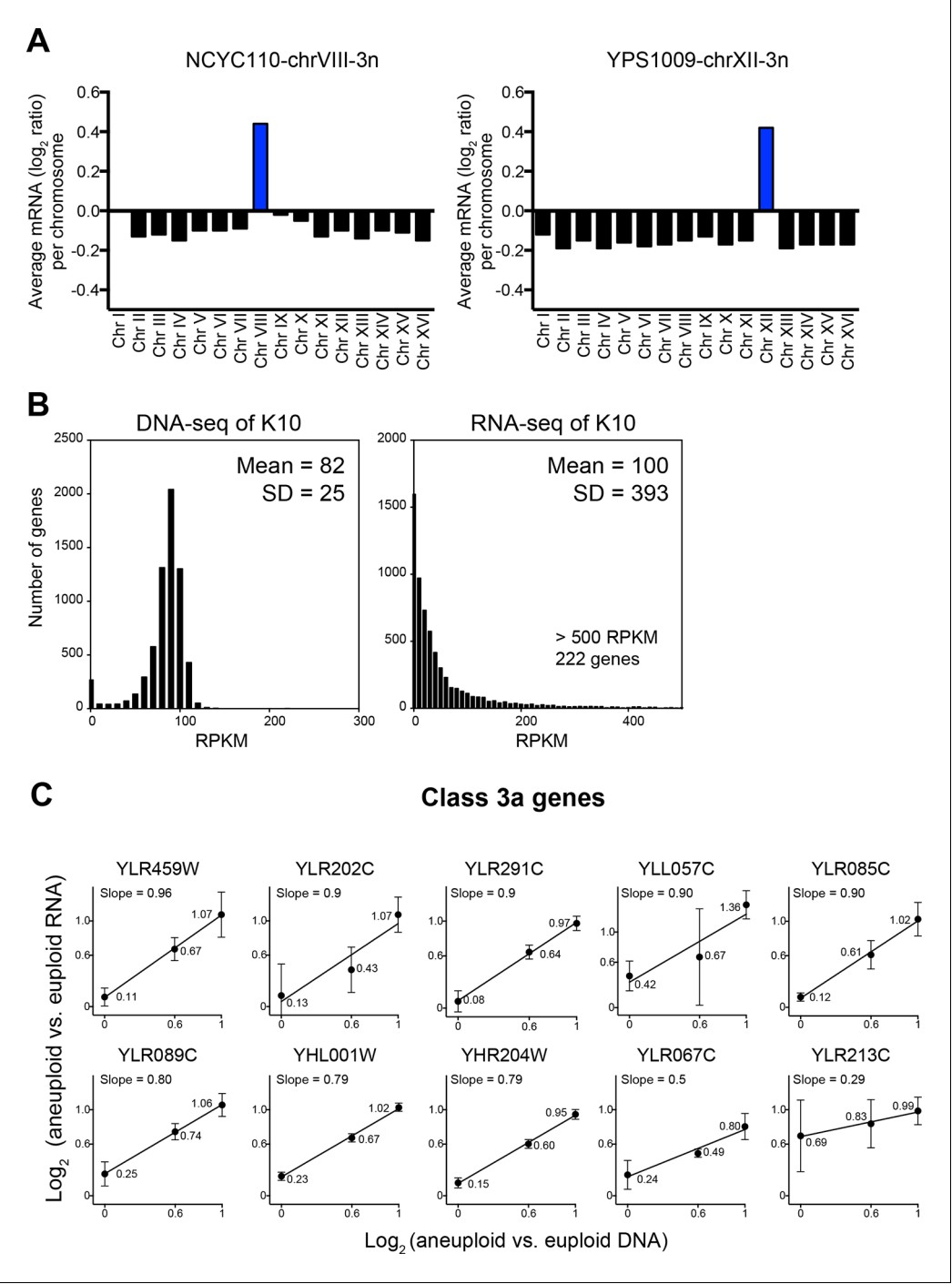

**Figure 8.** Evaluation of the analysis tools employed by *Hose et al. (2015)*. (**A**) RNA copy numbers averaged per chromosome of normalized RNA-seq data obtained by *Hose et al. (2015)*. Data provided by *Hose et al. (2015)*. (**B**) Standard deviations of RNA-seq data are greater than those of DNA-seq data. Histograms of DNA-seq RPKM and RNA-seq RPKM for strain K10 are shown. (**C**) Linear regression fits of RNA versus DNA copy number are shown for several genes identified as class 3a dosage compensated genes by *Hose et al. (2015)*. Eight genes from chromosome XII and two genes from chromosome VIII are shown. Average $log_2$ ratio of aneuploid vs. euploid RNA is shown. Error bars represent SD from three biological replicates.

# Materials and methods

## Karyotype heterogeneity analysis

We consider any chromosome whose copy number was significantly different from an integral value to be heterogeneous. To determine which chromosomes were significantly different than the nearest integer value, we used a one sample $t$ test using the copy number of each gene on the chromosome as the input which compares a distribution of values to an expected value and then corrected for multiple hypothesis testing. In strain YJM428, the expected value for chromosome III is 2 and the expected value for chromosome XV is 2. In strain Y2189, the expected value for chromosome I is 4, and for chromosomes IX and X is 3. In strain K1, the expected value for chromosomes I and VI is 2. In strains Y3 and Y6, the expected value for chromosome VIII is 3. In strain UC5, the expected value for chromosome I is 2. In strain CBS7960, the expected value for chromosomes I and III is 1. In strain WE372, the expected value for chromosome I is 3.

## Data processing

To avoid any discrepancies in data processing, Hose and coworkers kindly provided all the relative $\log_2$ ratios of the relative DNA copy number for all 47 wild strains and for the different panel of isogenic strains. In addition, Hose and coworkers kindly provided all gene expression data utilized in their manuscript. In addition, genome sequences for 16 distinct karyotypes (eight aneuploid and eight euploids) could be obtained from the NIH Sequence Read Archive (SRA) under accession SRP047341. Gene expression data could also be obtained from NIH GEO under accession GSE61532.

## Data normalization

$\log_2$ ratios provided by *Hose et al. (2015)* were normalized by centering the euploid chromosome ratios to 0. This was accomplished by calculating the mean of the $\log_2$ ratios of non-duplicated

**Table 2.** RNA copy number of aneuploid chromosomes in strain series NCYC110 and YPS1009. Analysis of genes encoded by chromosome VIII in strains NCYC110-2n, NCYC110-3n, NCYC110-4n (top) and encoded by chromosome XII in strains YPS1009-2n, YPS1009-3n, YPS1009-4n. One SD was used as a cutoff to identified genes with lower than expected RNA levels in each biological replicate. The "All 3 replicates" line represents genes whose RNA levels are reproducibly lower than expected in 3 RNA-seq experiments. Line "Both 3n and 4n" represent the number of genes whose RNA levels are lower than expected in trisomic and tetrasomic strains.

| NCYC110 | | | | | | | | | |
|---|---|---|---|---|---|---|---|---|---|
| | ChrVIII.2n-1 | ChrVIII.2n-2 | ChrVIII.2n-3 | ChrVIII.3n-1 | ChrVIII.3n-2 | ChrVIII.3n-3 | ChrVIII.4n-1 | ChrVIII.4n-2 | ChrVIII.4n-3 |
| Mean | 0.04 | -0.02 | 0.02 | 0.54 | 0.51 | 0.61 | 0.97 | 0.91 | 0.99 |
| Number of genes | 282 | 285 | 283 | 284 | 286 | 282 | 285 | 286 | 284 |
| SD | 0.53 | 0.56 | 0.54 | 0.51 | 0.50 | 0.54 | 0.57 | 0.55 | 0.56 |
| Mean - 1*SD | 23 | 18 | 20 | 29 | 25 | 22 | 24 | 33 | 29 |
| All 3 replicates | | | 3 | | | | | | 12 |
| Both 3n and 4n | | | 1 | | | | | | 2 |

| YPS1009 | | | | | | | | | |
|---|---|---|---|---|---|---|---|---|---|
| | Chr XII-2n-1 | Chr XII-2n-2 | Chr XII-2n-3 | Chr XII-3n-1 | Chr XII-3n-2 | Chr XII-3n-3 | Chr XII-4n-1 | Chr XII-4n-2 | Chr XII-4n-3 |
| Mean | 0.01 | 0.04 | 0.06 | 0.57 | 0.57 | 0.60 | 0.93 | 0.96 | 1.00 |
| Number of genes | 495 | 500 | 496 | 498 | 499 | 499 | 499 | 499 | 500 |
| SD | 0.41 | 0.52 | 0.46 | 0.46 | 0.47 | 0.51 | 0.65 | 0.66 | 0.66 |
| Mean - 1*SD | 42 | 56 | 36 | 47 | 50 | 46 | 46 | 52 | 45 |
| All 3 replicates | | | 8 | | | 15 | | | 17 |
| Both 3n and 4n | | | 3 | | | | | | 7 |

genes and subtracting this factor from all data points. Chromosome copy numbers were calculated by taking the average copy number of all genes within each chromosome. For diploids copy number equals $2*2$(log$_2$ ratio euploid vs. aneuploid), for haploids copy number equals $1*2$(log$_2$ ratio euploid vs. aneuploid).

Gene expression data of each aneuploid strain were compared to their reference genome as described in *Hose et al. (2015)*. Log$_2$ ratios of aneuploid/euploid genes were normalized to the euploid chromosomes as described above for DNA. RNA copy numbers per chromosome were calculated by averaging gene copy number of the genes within each chromosome.

## RNA and DNA distribution analysis of euploid and amplified genes

To analyze the distributions of euploid or amplified genes, DNA and RNA log$_2$ ratios of aneuploid/euploid we first calculated the distribution of the log$_2$ ratios binned by a value of 0.2. The frequency distributions were plotted and the data were fit to normal distribution utilizing PRISM software. Means, medians, SD, skewness and R$^2$ of the fits are reported in each figure. Gene expression data for disomes V and XVI (*Figure 3C*) were previously published in *Dephoure et al. (2014)*. Gene expression profiles were visualized with Treeview.

## Determination of false discovery rate

### Permutation

First, we needed to determine an effective cutoff to classify a gene as dosage compensated. We calculated the RNA/DNA ratio or slope of RNA versus DNA for all genes and binned the data (0.1 width bins in log space). For each bin, we then determined the percent of genes in that bin that were classified as dosage compensated by *Hose et al. (2015)*. Second, we randomized the data. We took the processed data (RNA/DNA) or raw data (RNA and DNA measurements, for slope analysis) and randomized the position of this information in the dataset. This decouples all the replicate measurements. *Hose et al. (2015)* supplied us with the RNA and DNA values for each gene and for each strain that they used to assess dosage compensation. Starting with this table as our input for randomization, we then calculated the RNA/DNA ratio for every replicate of every gene. We then permuted each column of the table (the replicates) independently and then calculated the average dosage compensation per 'gene' by averaging across the replicates; this is identical to how *Hose et al. (2015)* calculated dosage compensation from the unpermuted table.

If a subset of genes on a chromosome are compensated, as reported by *Hose et al. (2015)*, their average RNA/DNA ratios should appear as outliers on a distribution of RNA/DNA for a whole chromosome. Randomization of the RNA/DNA values before averaging will eliminate most of these outliers, as the outlying values will be most often average with non-outlying values; hence one should observe fewer genes that have large deviations from the mean. To assess this, we took all genes that *Hose et al. (2015)* had reported as dosage compensated. We took the distribution of RNA/DNA for these compensated genes and called this the observed or reported compensated distribution. The existence of true compensators would lead to significantly more genes in the compensated distribution than in the randomized distribution for a given dosage compensation range. This was not the case. Instead, the distributions were indistinguishable suggesting that the vast majority of genes reported as dosage compensated by *Hose et al. (2015)* is noise.

### Random sampling based on noise

Before calculating the false positive rate, one minor correction was needed. As the cutoff for calling a gene dosage compensated in *Hose et al. (2015)* did not take into account all the measurement noise we had to determine the effective cutoff used by *Hose et al. (2015)*.

While the RNA values were reported for each of the replicates, the DNA value was only reported as the mean of all measurements. This meant that calculating a SD based on the RNA/DNA ratios reported by *Hose et al. (2015)* would underestimate the true error of the measurement of dosage compensation and hence would give an artificially low false discovery rate. We turned to the sequencing data deposited with the paper, but the DNA data was only deposited for a subset of the strains. We therefore calculated the per gene DNA copy number error from the strains from which the replicates were deposited. From this, we found that the average standard deviation in DNA copy number was approximately 10%. For each dosage compensation value, we therefore randomly

sampled from a normal distribution with a SD of 10% and modified the dosage compensation value by this percentage (square root of squared sum of errors).

To determine the false discovery rate of this compensated distribution we compared the distribution of dosage compensated genes to the distribution of dosage compensation data from a distribution obtained by randomly sampling from a normal distribution with errors that came from a table of measurement errors. If we did not include the DNA error in this table of measurement estimates, the FDR rates dropped by about 10%. Thus, the vast majority of dosage compensated genes are most likely false positives irrespective of whether a correction was included or not.

## Acknowledgements

We are grateful to Audrey Gasch for providing data and analysis methods. This work was supported by the Richard and Susan Smith Family Foundation and the Searle Scholars Program to ET, and by the National Institute of Health (GM056800) to AA. AA is also an investigator of the Howard Hughes Medical Institute.

## Additional information

### Funding

| Funder | Author |
| --- | --- |
| Howard Hughes Medical Institute | Angelika Amon |
| Searle Scholars Program | Eduardo M Torres |

The funders had no role in study design, data collection and interpretation, or the decision to submit the work for publication.

### Author contributions

EMT, MS, AA, Conception and design, Analysis and interpretation of data, Drafting or revising the article

## Additional files

### Major datasets

The following previously published datasets were used:

| Author(s) | Year | Dataset title | Dataset URL | Database, license, and accessibility information |
| --- | --- | --- | --- | --- |
| Hose J, Yong CM, Sardi M, Wang Z, Newton MA, Gasch AP | 2015 | DNA Sequence | http://www.ncbi.nlm.nih.gov/sra/?term=SRP047341 | Publicly available at the NCBI Short Read Archive (Accession no: SRP047341). |
| Hose J, Yong CM, Sardi M, Wang Z, Newton MA, Gasch AP | 2015 | RNA Sequence | http://www.ncbi.nlm.nih.gov/geo/query/acc.cgi?acc=GSE61532 | Publicly available at the NCBI Gene Expression Omnibus (Accession no: GSE61532). |

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
