## [Decision Letter]

Thank you for submitting your work entitled "No evidence of widespread dosage compensation in wild *S. cerevisiae* strains" for consideration by *eLife*. Your article has been reviewed by four peer reviewers, and the evaluation has been overseen by a Reviewing Editor and Randy Schekman as the Senior Editor.

The reviewers have discussed the reviews with one another and the Reviewing editor has drafted this decision to help you prepare a revised submission.

Summary:

All the reviewers, as well as the Reviewing Editor, were entirely convinced that a revised and carefully re-written version of your manuscript should be publicly released in *eLife*. Major concerns centered around the overall tone being too aggressive, many analyses possibly being too stringent, and an unbalanced consideration of all possible explanations, particularly given how nascent our understanding as a field is of dosage compensation. It was also discussed that the authors need to describe the methods used in a more detailed manner.

All four reviewers brought up a highly compelling set of additional analysis and interpretation concerns to be addressed, and are therefore appended to this letter. A revision suitable for acceptance will not require any additional experimental data; however, to publish will require new analyses, corrected figures, a balanced Discussion, and substantial reworking of the text and argument structure.

Essential revisions:

1) Methods: Every analysis should be described in a detailed Methods section, with subsections clearly annotated and referenced in the main text by occurrence in the figures and in the Results section.

2) Soften tone: As one reviewer eloquently said: "The tone throughout the article is rather strident and bordering on confrontational. It would be wise for the sake of rational open discussion to soften some of the more forceful statements." The overly confident or aggressive text must be adjusted before acceptance. In some sections of the manuscript, the authors’ interpretations were simplified or inappropriate, and exceptionally strict analysis cut offs may bias Torres' interpretations. Additional tonal comments found in the reviews below should not be neglected. Importantly, the title and section headers will have to be toned down as well.

3) Restructure: A careful introduction to what is meant by 'dosage compensation' must be laid out clearly in the Introduction. Sections should start with a paragraph laying out the reasoning that leads to why each analysis was performed (see for instance Reviewer 3 point 5). See also the Reviewing Editor’s comments below.

4) Aneuploidy versus instability: Many reviewers noted that Torres' equating aneuploidy and instability was not appropriate; this must be corrected throughout.

*Reviewer #1:*

The Torres et al. manuscript re-examines the Hose et al. (*eLife* 2015) data to evaluate dosage compensation in wild yeast strains. At the heart of the Torres manuscript is whether the data of Hose et al. truly support dosage compensation. Their analysis is thorough, careful, and convincing, but the methods need to be better described.

1) They argue that the presence of non-integer DNA copy number states implies the strains are highly heterogeneous and unstable. But this is methodologically poorly described. The methods argue that chromosome copy number is calculated by taking the average number of all genes within each chromosome. So one assumes that non-integer is an average substantially different from an integer, but what variance is acceptable? The copy number plots (Figure 1, Figure 2, Figure 4) show considerable variability from point to point, depending on strain; for example Figure 1 NCYC110 is tightly distributed around 2 DNA copies whereas YPS1009 seems more variable. Is this variability taken into account? Is it influenced by depth? Means are more sensitive to outliers than the median – would you see a similar result using the median, or (in the case of numerous aneuploid chromosomes) is the median too skewed upward?

Furthermore, it is unclear that they can infer instability rather than simply heterogeneity, the exception being the one obvious case (K1) where the DNA and RNA copy numbers vary. That said, a devil's advocate argument is that K1 shows differences between DNA and RNA assays (specifically on chromosome VI) because of "dosage compensation". While highly unlikely (seeing how there are other aneuploid chromosomes that are not compensated), it seems an important point given the final overall findings of the paper.

2) The methodologies for assessing false positives are poorly described. Both the randomization and the error/noise distribution approach are only vaguely described. When the ratios were permuted relative to the gene list, what was then done to assess dosage compensation – they say "the method of Hose et al. (see Methods)” but in the Methods it is not described. (The false discovery rate analysis is described but that is used later in the paper). What does "not reported independently" (in this same section) mean? How does using the "lowest average chromosome-wide DNA error" bias your result? How exactly were errors combined?

3) The heart of the paper is the issue of how to assess dosage compensation. They miss an opportunity in the Discussion to discuss both the key points of "dosage compensation" as a concept (distinct from what they refer to as "transcriptional response to aneuploidy”). Likewise there are issues when assessing aneuploidy numbers – the constant need for a frame of reference and this issue of expectation (i.e. RNA copy numbers will vary relative to DNA simply because of expression levels being variable). This would help the reader to understand why skew is expected in the distribution if dosage compensation is present.

*Reviewer #2:*

Amon and coworkers re-analyze data previously obtained by Hose et al. (2015). This re-evaluation identifies several flaws in the original analysis and yields a completely opposite conclusion; namely that there is no sign of widespread dosage compensation in (aneuploid/polyploid) feral *S. cerevisiae* strains. As far as I can see, the re-analysis is technically sound and I especially commend the authors for the permutation test applied to the RNA/DNA ratio for each gene.

I therefore recommend publication of this paper in *eLife*, even though I do have a few suggestions (below).

1) Most importantly, I would suggest giving A. Gash and her team the opportunity to co-publish their response to this new paper together with the publication of the new paper. I think it would be interesting to know what the original authors think about this re-evaluation, and it also seems courteous to offer this possibility to Dr. Gash. I also feel that it is important to give the opportunity to have the response published together (at the same time) with this new paper.

2) It would be interesting to expand the short Discussion section to further highlight that non-laboratory *S. cerevisiae* strains harbour natural copy number variants. The sentence "The fact that strains YJM428, Y2189, K1, UC5, Y3, Y6 and CBS7960 are unstable also means that these strains are less fit than euploid strains" might benefit from a more elaborate discussion to help the reader understand the rationale behind this argument, and to discuss literature showing that several experimental evolution experiments have identified transient aneuploidies as a common but potentially suboptimal solution to overcome harsh conditions.

3) The authors use non-integer changes in DNA read depth as evidence that a given strain is unstable. I see the logic behind this reasoning, but since this result directly contradicts previously published results, and since it is of great importance to the broad community working with (industrial and feral) *S. cerevisiae* yeasts, one has to be sure that there are no (cryptic) technical reasons for the non-integer changes, even if this scenario appears unlikely. If the strains are indeed so unstable, one would expect that analyses (whole-genome-sequencing) of different single colonies yield (very) different outcomes. Is this the case? Another, more elegant approach would be to investigate CNVs in single cells in a population, but this might be technically more challenging…

*Reviewer #3:*

This manuscript is a report that is a rebuttal directed at a prior *eLife* paper by Gasch and her group (Hose et al, 2015). The paper by Gasch reports dosage compensation at the transcriptional level in aneuploid wild yeast. The main focus of the reviewed manuscript is to demonstrate that the observed dosage compensation reflects shortcomings of the analysis. Science should foster open disputation of controversial topics. As the field of aneuploidy is rather new and evolving, it is essential to address the disagreements early on and thoroughly. Torres et al. specifically mention the following issues: the used strains may be unstable, the normalization is suboptimal there is no test for significance of the number of dosage compensated genes and its correction for multiple testing the standard deviation of the DNA sequencing data is used as a cut off for determining dosage compensated genes on RNA sequencing data and the three-point analysis to identify dosage compensated genes is too limited and therefore prone to noise-related errors. Since the analysis by Hose et al. has some flaws, this referee feels that the manuscript by Torres et al. should be made public. However, there are several important points that should be addressed.

1) First, the authors make the point that the wild aneuploid yeast might be chromosomally unstable. Although this is probably true, it is not possible to conclude based on the presented data – the authors say as an example that the RNA and DNA levels are very different for K1 strain in Figure 1 in Hose et al.; however, this reviewer can see only DNA levels in 1A. The fact that there are non-integer DNA copy numbers means strictly speaking only that the strain is heterogeneous.

2) In the second paragraph of "Evaluation of the analysis methods employed by Hose et al.", the authors state: "Most normalization protocols do not take into account that aneuploid strains harbor a different total number of genes than euploid strains". This statement by Torres et al. is incorrect. If something, they contain more copies of genes. Also, Torres et al. mention that the data has to be manually corrected, however, there is no description how they envision to do this nor is it obvious whether they performed it on the data from Hose et al. or not. Here, a description should be added at least in the Methods description. Next, Torres et al. state that "the data used for analysis by Hose et al. (2015) deviate from the actual expression values..."; here the "actual expression values" should be those that Torres et al. calculated. In fact, there is no way to know the "actual expression values"; the approach is to just try to normalize the data and hope that it reflects the reality.

3) To straighten their point, the authors should show at least one volcano plot of the 838 compensated genes (identified by Hose et al.) and their calculated FDRs.

4) It would be useful for the reader if the authors would add the 2SD RNA and 2SD DNA to Table 1 for comparison of the variance and further substantiate the incorrect use of the 2SD DNA as a cutoff in Hose et al.

5) For the understanding of the discussed issues it might be useful to restructure the manuscript in a way that each of the three paragraphs showing that there is no dosage compensation includes the corresponding critical points of the analysis by Hose et al. For each of the three subchapters first the criticism should be raised, followed by their re-analysis and their own analysis. It feels much more logical to explain first the criticism and then show the new results than to list first everything that is wrong and only then show the reasons.

*Reviewer #4:*

This manuscript is a very thorough reanalysis of a recent paper on aneuploidy and wild strains recently published in *eLife* (Hose et al.). This paper reaches quite different conclusions than Hose et al. and instead shows a lack of evidence for what Hose et al. call dosage compensation. I found the Torres et al. analysis pretty compelling, and I was rather surprised that the original paper did not include similar analyses. Some of the identified mistakes include errors in normalization, apparent copy number variants in the strains used as euploid controls, and discrepancies between RNA and DNA for some strains. Overall I was convinced that there are significant issues with the Hose paper that were masked by the data presentation and analysis methods used.

I was particularly convinced by the demonstration that the distribution of gene expression of genes on euploid and amplified chromosomes was identical. One could argue whether the distributions of DNA and RNA should be compared (even given the vastly different dynamic ranges), or whether the exact cutoffs employed are too stringent or not stringent enough, but any model of "dosage compensation" seems like it should have genes behaving differently when they are at elevated dosage. This does not appear to be the case.

That said, I did find some of the reanalysis a little too demanding: buffering could still be present but below the 2 SD thresholds used throughout. That's one difficulty of working with single increment dosage effects, particularly in diploids: the fold changes expected are far less than the cutoffs generally employed for expression analysis. While in bulk it's clear that the average expression change scales with copy number, individual genes may not. The MLR analysis from the dosage series strains seems like the best way to detect these potentially subtle effects, though the false positive analysis was fairly convincing that even this analysis fell short in Hose et al.

To make the paper even more compelling, I would like to see some positive controls. That is, what would genes look like that truly do have some buffering against dosage? Can they ever be detected using the methods of either this paper or the Hose et al. one? The Introduction mentions ribosomal genes and histones, for example, so it would be interesting to see how they behave. It could be that detecting such subtle effects is currently beyond the abilities of current RNA measurement technologies.

In addition, the main argument of the Hose paper seems to be a difference between lab strains and natural isolates. A more direct comparison between the data from different strains in this paper would help compare these directly and demonstrate whether they are in fact showing identical patterns.

Reviewing Editor's comments:

The following editorial revisions are single examples of the type that must be made to the manuscript's tone, listed in order of occurrence in the manuscript. Re-review will focus on a detailed list of revisions, if edits are still needed.

Title

New title: “No current evidence for widespread dosage compensation in yeast”.

Abstract alterations:

“mostly lead to an according change in gene expression.” This is a nonsensical sentence, likely a typo.

“gene expression is not observed in wild…” to "gene expression can be violated in wild…".

“dosage compensation occurs neither in laboratory strains nor natural…” to "dosage compensation remains unproven in laboratory strains and natural variants."

Results headers:

All these should be re-worded to report results, not conclusions. For instance:

"Many wild yeast strains have unstable karyotypes." As the reviewers indicated, this is just one of the potential interpretations of the CNV data.

"No dosage compensation in…" (all sections). These are extrapolative and interpretive statements that should be toned down considerably.

Section titled "No dosage compensation in wild isolated YJM428…" should probably be split into two sections, based on the fact that the six yeast lines listed are explored as falling under two different explanations.

---

## [Author Response]

*Essential revisions:*

1) Methods: Every analysis should be described in a detailed Methods section, with subsections clearly annotated and referenced in the main text by occurrence in the figures and in the Results section.

The Methods have been extended and subsections were added.

2) Soften tone: As one reviewer eloquently said: "The tone throughout the article is rather strident and bordering on confrontational. It would be wise for the sake of rational open discussion to soften some of the more forceful statements." The overly confident or aggressive text must be adjusted before acceptance. In some sections of the manuscript, the authors’ interpretations were simplified or inappropriate, and exceptionally strict analysis cut offs may bias Torres' interpretations.

We had not intended to sound aggressive. We thought we just stated the facts but clearly the reviewers thought otherwise. We have tried to soften the tone and hope to have done so, but given that we did not think we sounded aggressive when we first wrote the paper we ask the editor to ensure that we softened the tone appropriately.

We would also like to address the concern that the cutoffs we used to evaluate the Hose et al. (2015) analysis were too stringent. We have calculated the false discovery rates for cutoffs ranging from 0.1 – 2SDs. The FDR is 93% or more in all instances. We have included this analysis in the revised manuscript.

Additional tonal comments found in the reviews below should not be neglected. Importantly, the title and section headers will have to be toned down as well.

The title and section headers have been changed according to the reviewing editor’s suggestions.

*3) Restructure: A careful introduction to what is meant by 'dosage compensation' must be laid out clearly in the Introduction. Sections should start with a paragraph laying out the reasoning that leads to why each analysis was performed (see for instance Reviewer 3 point 5). See also the Reviewing Editor’s comments below.*

We have extended the description and definition of dosage compensation in the Introduction. Furthermore, we have restructured the manuscript. The Results section of the manuscript is now divided into three sections. The first discusses the heterogeneity of the strains Hose et al. (2015) analyzed and how this affects the subsequent analyses. Section 2 describes our analyses of the gene expression study conducted by Hose et al. (2015). Section three discusses why Hose et al. (2015) arrived at the wrong conclusions. All three sections are prefaced by a paragraph explaining what has been done before and why we did what we did.

*4) Aneuploidy versus instability: Many reviewers noted that Torres' equating aneuploidy and instability was not appropriate; this must be corrected throughout.*

This has been corrected throughout the text.

*Reviewer #1:*

The Torres et al. manuscript re-examines the Hose et al. (eLife 2015) data to evaluate dosage compensation in wild yeast strains. At the heart of the Torres manuscript is whether the data of Hose et al. truly support dosage compensation. Their analysis is thorough, careful, and convincing, but the methods need to be better described.1) They argue that the presence of non-integer DNA copy number states implies the strains are highly heterogeneous and unstable. But this is methodologically poorly described. The methods argue that chromosome copy number is calculated by taking the average number of all genes within each chromosome. So one assumes that non-integer is an average substantially different from an integer, but what variance is acceptable?

We consider any chromosome whose copy number was significantly different from an integral value to be heterogeneous. To determine which chromosome were significantly different than the nearest integer value, we used a one sample t-test using the copy number of each gene on the chromosome as the input which compares a distribution of values to an expected value and then corrected for multiple hypothesis testing. Chromosomes in Figure 1 and Figure 2 labeled with asterisks show significant deviations from the expected values.

*The copy number plots (Figure 1, Figure 2, Figure 4) show considerable variability from point to point, depending on strain; for example Figure 1 NCYC110 is tightly distributed around 2 DNA copies whereas YPS1009 seems more variable. Is this variability taken into account? Is it influenced by depth? Means are more sensitive to outliers than the median* – *would you see a similar result using the median, or (in the case of numerous aneuploid chromosomes) is the median too skewed upward?*

All medians are identical to the means. This has been added to the figure legends. The distributions are not skewed.

*Furthermore, it is unclear that they can infer instability rather than simply heterogeneity, the exception being the one obvious case (K1) where the DNA and RNA copy numbers vary. That said, a devil's advocate argument is that K1 shows differences between DNA and RNA assays (specifically on chromosome VI) because of "dosage compensation". While highly unlikely (seeing how there are other aneuploid chromosomes that are not compensated), it seems an important point given the final overall findings of the paper.*

The strains analyzed by Hose et al. (2015) were all obtained from a single colony. Because of this, heterogeneity observed in the strains can only arise through genomic instability.

Because the RNA and DNA samples were not collected from the same culture, any unusual divergence between RNA and DNA would be additional evidence for instability. K1 and Y2189 both show unexpected differences between the RNA and DNA. For example, strain Y2189 shows amplification of chromosome IV at the RNA level but not DNA level (Figure 1). In summary, the non-integer values of DNA copy number are the strongest sign of instability, the RNA/DNA discrepancy just supports this conclusion.

*2) The methodologies for assessing false positives are poorly described. Both the randomization and the error/noise distribution approach are only vaguely described. When the ratios were permuted relative to the gene list, what was then done to assess dosage compensation* – *they say "the method of Hose et al. (see Methods)” but in the Methods it is not described. (The false discovery rate analysis is described but that is used later in the paper). What does "not reported independently" (in this same section) mean? How does using the "lowest average chromosome-wide DNA error" bias your result? How exactly were errors combined?*

The following description was added to the Methods section:

“Permutation:

First we needed to determine an effective cut-off to classify a gene as dosage compensated. […] This was not the case. Instead the distributions were indistinguishable suggesting that the vast majority of genes reported as dosage compensated was simply noise.

Random sampling based on noise:

Before calculating the false positive rate, one minor correction was needed. As the cutoff for calling a gene dosage compensated in Hose et al. (2015) did not take into account all the measurement noise we had to determine the effective cutoff used by Hose et al. (2015). […] Thus, the vast majority of dosage compensated genes are most likely false positives irrespective of whether a correction was included or not.”

*3) The heart of the paper is the issue of how to assess dosage compensation. They miss an opportunity in the Discussion to discuss both the key points of "dosage compensation" as a concept (distinct from what they refer to as "transcriptional response to aneuploidy”). Likewise there are issues when assessing aneuploidy numbers – the constant need for a frame of reference and this issue of expectation (i.e. RNA copy numbers will vary relative to DNA simply because of expression levels being variable). This would help the reader to understand why skew is expected in the distribution if dosage compensation is present.*

We have added a discussion of how a transcriptional response versus dosage compensation would be borne out by the data in the Introduction and added a graph that shows how distributions would be skewed if dosage compensation occurs (Figure 7).

*Reviewer #2:*

*Amon and coworkers re-analyze data previously obtained by Hose et al. (2015). This re-evaluation identifies several flaws in the original analysis and yields a completely opposite conclusion; namely that there is no sign of widespread dosage compensation in (aneuploid/polyploid) feral S. cerevisiae strains. As far as I can see, the re-analysis is technically sound and I especially commend the authors for the permutation test applied to the RNA/DNA ratio for each gene. I therefore recommend publication of this paper in eLife, even though I do have a few suggestions (below). 1) Most importantly, I would suggest giving A. Gash and her team the opportunity to co-publish their response to this new paper together with the publication of the new paper. I think it would be interesting to know what the original authors think about this re-evaluation, and it also seems courteous to offer this possibility to Dr. Gash. I also feel that it is important to give the opportunity to have the response published together (at the same time) with this new paper.*

In the spirit of truth seeking and collegiality, we had reached out to Dr. Gasch before submitting this manuscript. We had numerous e-mail exchanges and a phone conversation in the hope to reach a consensus. Throughout these conversations Dr. Gasch indicated that she is standing by her data and interpretations but we were not able to understand her reasoning. As we were not able to reach a consensus in these conversations we submitted this paper to *eLife* for publication. At the same time we sent her a copy of the manuscript.

*2) It would be interesting to expand the short* Discussion section *to further highlight that non-laboratory S. cerevisiae strains harbour natural copy number variants. The sentence "The fact that strains YJM428, Y2189, K1, UC5, Y3, Y6 and CBS7960 are unstable also means that these strains are less fit than euploid strains" might benefit from a more elaborate discussion to help the reader understand the rationale behind this argument, and to discuss literature showing that several experimental evolution experiments have identified transient aneuploidies as a common but potentially suboptimal solution to overcome harsh conditions.*

Based on the recommendation by Reviewer 1 we have removed the discussion of the consequences of chromosome instability on cellular fitness. However, we should emphasize that the assumption that these wild-strains are naturally aneuploid is likely to be incorrect. The observation that the strains are so unstable when grown in the laboratory raises the distinct possibility that the aneuploidies observed in these strains are a consequence of culturing the natural variants under laboratory conditions. We have added these thoughts to the paper.

*3) The authors use non-integer changes in DNA read depth as evidence that a given strain is unstable. I see the logic behind this reasoning, but since this result directly contradicts previously published results, and since it is of great importance to the broad community working with (industrial and feral) S. cerevisiae yeasts, one has to be sure that there are no (cryptic) technical reasons for the non-integer changes, even if this scenario appears unlikely. If the strains are indeed so unstable, one would expect that analyses (whole-genome-sequencing) of different single colonies yield (very) different outcomes. Is this the case? Another, more elegant approach would be to investigate CNVs in single cells in a population, but this might be technically more challenging…*

Technical problems are not the reason for the results we obtained. A small number of the strains that Hose et al. (2015) have characterized are stable and harbor homogenous karyotypes as judged by whole integer changes in chromosome copy number.

The disparity between RNA and DNA measurements is further evidence that these strains are unstable. Because RNA and DNA were not isolated from the same sample, any unusual divergence between RNA and DNA is additional evidence for instability. K1 and Y2189 both show unexpected differences between the RNA and DNA, e.g. the average copy number of chr III in strain K1 is 4 (log_2_ ratio = 0.98) while the average increase in gene expression is 3-fold (log_2_ ratio = 0.63).

*Reviewer #3:*

This manuscript is a report that is a rebuttal directed at a prior eLife paper by Gasch and her group (Hose et al, 2015). The paper by Gasch reports dosage compensation at the transcriptional level in aneuploid wild yeast. The main focus of the reviewed manuscript is to demonstrate that the observed dosage compensation reflects shortcomings of the analysis. Science should foster open disputation of controversial topics. As the field of aneuploidy is rather new and evolving, it is essential to address the disagreements early on and thoroughly. Torres et al. specifically mention the following issues: the used strains may be unstable, the normalization is suboptimal there is no test for significance of the number of dosage compensated genes and its correction for multiple testing the standard deviation of the DNA sequencing data is used as a cut off for determining dosage compensated genes on RNA sequencing data and the three-point analysis to identify dosage compensated genes is too limited and therefore prone to noise-related errors. Since the analysis by Hose et al. has some flaws, this referee feels that the manuscript by Torres et al. should be made public. However, there are several important points that should be addressed.

*1) First, the authors make the point that the wild aneuploid yeast might be chromosomally unstable. Although this is probably true, it is not possible to conclude based on the presented data*– *the authors say as an example that the RNA and DNA levels are very different for K1 strain in Figure 1 in Hose et al; however, this reviewer can see only DNA levels in 1A. The fact that there are non-integer DNA copy numbers means strictly speaking only that the strain is heterogenous.*

RNA and DNA levels for the heterogeneous strains are shown in Figure 1. Importantly, the strains analyzed by Hose et al. (2015) were all obtained from a single colony. Because of this, the heterogeneity observed in the strains can only arise through genome instability. Non-integer values of DNA copy number therefore are a strong sign of instability; the RNA/DNA discrepancy supports this conclusion.

2) In the second paragraph of "Evaluation of the analysis methods employed by Hose et al.", the authors state: "Most normalization protocols do not take into account that aneuploid strains harbor a different total number of genes than euploid strains". This statement by Torres et al. is incorrect. If something, they contain more copies of genes.

It depends, when analyzing monosomies, the number of genes in the aneuploid strain would be smaller. In the case of the strains analyzed by Hose et al. (2015) the reviewer is correct, the strains contained more gene copies but given that we wanted to make a more general statement we wish to keep the phrasing as is.

Also, Torres et al. mention that the data has to be manually corrected, however, there is no description how they envision to do this nor is it obvious whether they performed it on the data from Hose et al. or not. Here, a description should be added at least in the Methods description.

We manually corrected the data for our analysis. A description has been added to the Methods.

Next, Torres et al. state that "the data used for analysis by Hose et al. (2015) deviate from the actual expression values..."; here the "actual expression values" should be those that Torres et al. calculated. In fact, there is no way to know the "actual expression values"; the approach is to just try to normalize the data and hope that it reflects the reality.

We meant correctly normalized values. This has been corrected.

*3) To straighten their point, the authors should show at least one volcano plot of the 838 compensated genes (identified by Hose et al.) and their calculated FDRs.*

Volcano blots are only useful if outliers exist in the data set that can be statistically identified. Because of the extremely high false discovery rate we cannot call any outliers with confidence. To assess whether cut-off stringency affects the results we examined the FDR across a wide range of cut-offs, 0.1 SD – 2SDs. The FDR is 93% or more in all instances. We have included this analysis in the revised manuscript.

*4) It would be useful for the reader if the authors would add the 2SD RNA and 2SD DNA to Table 1 for comparison of the variance and further substantiate the incorrect use of the 2SD DNA as a cutoff in Hose et al.*

Data have been added.

*5) For the understanding of the discussed issues it might be useful to restructure the manuscript in a way that each of the three paragraphs showing that there is no dosage compensation includes the corresponding critical points of the analysis by Hose et al. For each of the three subchapters first the criticism should be raised, followed by their re-analysis and their own analysis. It feels much more logical to explain first the criticism and then show the new results than to list first everything that is wrong and only then show the reasons.*

We thought about this suggestion but prefer to keep the structure as is. It gets very confusing to discuss the shortcomings of each experiment first and then show the correct analysis because every data set presented by Hose et al. (2015) suffers from the same shortcomings. We therefore kept the structure as is and added some additional explanations in each section.

*Reviewer #4:*

This manuscript is a very thorough reanalysis of a recent paper on aneuploidy and wild strains recently published in eLife (Hose et al.). This paper reaches quite different conclusions than Hose et al. and instead shows a lack of evidence for what Hose et al. call dosage compensation. I found the Torres et al. analysis pretty compelling, and I was rather surprised that the original paper did not include similar analyses. Some of the identified mistakes include errors in normalization, apparent copy number variants in the strains used as euploid controls, and discrepancies between RNA and DNA for some strains. Overall I was convinced that there are significant issues with the Hose paper that were masked by the data presentation and analysis methods used. I was particularly convinced by the demonstration that the distribution of gene expression of genes on euploid and amplified chromosomes was identical. One could argue whether the distributions of DNA and RNA should be compared (even given the vastly different dynamic ranges), or whether the exact cutoffs employed are too stringent or not stringent enough, but any model of "dosage compensation" seems like it should have genes behaving differently when they are at elevated dosage. This does not appear to be the case. That said, I did find some of the reanalysis a little too demanding: buffering could still be present but below the 2 SD thresholds used throughout.

We also examined the Hose et al. (2015) data using 1 SD as a cutoff. This analysis is shown in Table 2. We should further add that RNA-seq alone is not sufficient to identify single genes whose expression levels are partially dosage compensated. The reason for this is that the variance of the distributions of the genome-wide RNA measurements are greater than one would expect if gene expression is lower than 50% (SD of log_2_ ratios = 0.6 which represents 1.5 fold changes). Quantitative gene centricanalyses are required to identified genes that are significantly attenuated and do not increase in expression with increases in copy number.

*That's one difficulty of working with single increment dosage effects, particularly in diploids: the fold changes expected are far less than the cutoffs generally employed for expression analysis. While in bulk it's clear that the average expression change scales with copy number, individual genes may not. The MLR analysis from the dosage series strains seems like the best way to detect these potentially subtle effects, though the false positive analysis was fairly convincing that even this analysis fell short in Hose et al.*

We completely agree with this assessment. It is very difficult to detect potential dosage compensation of individual genes. Chromosome-wide, such phenomena can be detected but for single genes all methods, including the one developed by Hose et al. (2015) are confounded by high false positive rates.

*To make the paper even more compelling, I would like to see some positive controls. That is, what would genes look like that truly do have some buffering against dosage? Can they ever be detected using the methods of either this paper or the Hose et al. one? The Introduction mentions ribosomal genes and histones, for example, so it would be interesting to see how they behave. It could be that detecting such subtle effects is currently beyond the abilities of current RNA measurement technologies.*

We have previously shown that the method we employed to analyze the data by Hose et al. (2015) can identify dosage compensation. In Dephoure et al. (2014) we showed that ribosomal protein levels are significantly attenuated in disomic laboratory strains. The distributions of these expression levels exhibit significant negative skew, as one would expect from gene products that are dosage compensated. We are now referring to this study in the text.

*In addition, the main argument of the Hose paper seems to be a difference between lab strains and natural isolates. A more direct comparison between the data from different strains in this paper would help compare these directly and demonstrate whether they are in fact showing identical patterns.*

Data for laboratory strains disomic for chromosome V or XVI have been added in Figure 3.

Reviewing Editor's comments:

The following editorial revisions are single examples of the type that must be made to the manuscript's tone, listed in order of occurrence in the manuscript. Re-review will focus on a detailed list of revisions, if edits are still needed. Title New title: “No current evidence for widespread dosage compensation in yeast”.

Done.

*Abstract alterations:*

“mostly lead to an according change in gene expression.” This is a nonsensical sentence, likely a typo.

“gene expression is not observed in wild…” to "gene expression can be violated in wild…".

“dosage compensation occurs neither in laboratory strains nor natural…” to "dosage compensation remains unproven in laboratory strains and natural variants."

Results headers:All these should be re-worded to report results, not conclusions. For instance:

"Many wild yeast strains have unstable karyotypes." As the reviewers indicated, this is just one of the potential interpretations of the CNV data.

"No dosage compensation in…" (all sections). These are extrapolative and interpretive statements that should be toned down considerably.Section titled "No dosage compensation in wild isolated YJM428…" should probably be split into two sections, based on the fact that the six yeast lines listed are explored as falling under two different explanations.

All other editorial suggestions have been incorporated.